# A Novel Rail Damage Fault Detection Method for High-Speed Railway

**DOI:** 10.3390/s25103063

**Published:** 2025-05-13

**Authors:** Yu Wang, Bingrong Miao, Ying Zhang, Zhong Huang, Songyuan Xu

**Affiliations:** National Key Laboratory of Rail Transit Vehicle System, Southwest Jiaotong University, Chengdu 610032, China; yu2022wang@163.com (Y.W.);

**Keywords:** steel rail, high speed railway, robots, fault detection

## Abstract

With the vigorous development of speedy railway technology, steel rails, as an important structural infrastructure in speedy railways, play a crucial role in ensuring the safety of the entire speedy railway operation. A brand-new type of speedy rail inspection robot and its fault detection method are proposed to solve a number of problems, such as the difficulty and low accuracy of real-time online detection of rail defects and damage in speedy railways. The brand-new rail inspection robot is driven by two drive wheels and adopts a standard rail gauge of 1435 mm, which ensures its speedy and smooth operation on the track as well as accurate motion posture information. Firstly, 12 common types of surface damage of the rail head were analyzed and classified into five categories based on their damage characteristics. The motion state of the rail inspection robot under the five types of surface damage of the rail head was analyzed and subjected to kinematic analysis. This study analyzed the relationship between the distinctive types of damage and the motion posture of the robot during the inspection of the five types of damage. Finally, experimental tests were conducted, and it was found that the robot’s motion posture would undergo sudden changes when inspecting distinctive types of injuries; the highest error rate was 3%. The effectiveness of this method was verified through experiments, and the proposed new track detection robot can greatly improve the track detection efficiency of high-speed railways and has specific academic research value and practical application value.

## 1. Introduction

With the promulgation and implementation of the “Outline for Building a sturdy Transportation Country”, the railway system is an important component of a sturdy transportation country and a modern comprehensive transportation system, and railway technology and equipment have experienced rapid development. In particular, with the implementation of the national “Belt and Road” strategy and the deployment of the “eight vertical and eight horizontal” high-speed railway network [1], China’s railways have developed rapidly. The use of rail transit using eddy current testing technology to manually push the detection vehicle can be traced back to ancient Greece in the 6th century BC at the earliest [2]. For example, Liu Pingxin [3] from Shanghai Jiao Tong University proposed a method for detecting multiple rail cracks using guided waves based on deep learning, which is used for the rapid detection of rail surface damage. The quantitative evaluation method of rolling contact fatigue damage based on the wedge volume/cross-sectional area of crack damage, proposed by Xie Yulong [4] from Southwest Jiaotong University, quantitatively evaluates fatigue crack damage through a reasonable evaluation method. It can effectively assess the damage degree of rolling contact fatigue cracks and provide theoretical guidance for the maintenance of rolling element wear parts. In the previous century, due to the advantages of both large capacity and low transportation costs, railway transportation flourished, and the complete length of railway lines worldwide rapidly increased and exceeded 1.2 million kilometers, undertaking approximately 70% of the complete global freight volume and 50% of the passenger volume. With the introduction of electric locomotives and the demand for speedy railway transportation, speedy railway lines have become an important direction for countries around the world to vigorously develop rail transit. Ruipeng Gao et al. [5] analyzed the surface crack damage of the rail by analyzing the relationship between the fatigue crack propagation process of the rail and the number of cyclic loads. Xuefeng Ni et al. [6] proposed a novel algorithm for detecting rail surface defects based on partitioned edge features (PEFs) to address the identification and detection of rail surface damage. With the development of railway technology and the continuous expansion of the railway network, the carrying speed, load, and frequency of railway vehicles are constantly increasing. As an important component of railways, the surface damage defects of steel rails are also increasing. The healthy operation of steel rails has received more and more attention from researchers. Most of the damage to steel rails, such as cracking of the rail head, peeling of the rail surface, collapse of the rail surface, rail fracture, and rail head damage [7], are related to fatiguing cracks. Real-time health monitoring of rail surface damage, including identification and detection of rail surface cracked damage, plays a crucial role in the normal and healthy operation of railways and is also a key technical point in the intelligent application of railways.

During operation, steel rails often experience head wear and various rolling contact fatigue damages under alternating loads. The main forms of rail damage are as follows: peeling and crushing of rail head material, rail side wear, rail corrugation, welding joint damage, and fracture caused by rolling contact fatigue cracks. Mariusz Nieniewski et al. [8] created a system that utilizes morphological operations to detect rail defects and extract their shapes, achieving the detection of surface damage defects on the rails. Xiating Jin et al. [9] created a method of directly performing parameter learning in the Expectation-Maximization (EM) algorithm. At the same time, they adopted the Faster Region-based Convolutional Neural Network (Faster RCNN) with a parallel structure to achieve the target location of the damage on the rail surface, thus improving the ability to detect rail surface defects.

At present, the methods for identifying and detecting surface damage on steel rails are mainly divided into contact and non-contact methods according to the detection method. The contact detection method mainly uses some contact sensors or physical auxiliary components to collect damage signals on the surface of the steel rail, further process the damage signals, and finally determine the damage defects [10]. Representative testing methods include ultrasonic testing, magnetic particle testing, eddy current testing, magnetic leakage testing, radiographic testing, track circuit testing, acoustic emission technology, and ultrasonic guided wave testing technology [11]. The principle of ultrasonic testing is relatively simple, which is to emit ultrasonic pulse signals into the steel rail and determine the integrity of the steel rail based on parameters such as the amplitude and transmission time of the reflected and transmitted pulses. The main excitation methods of ultrasound include piezoelectric ultrasound, electromagnetic ultrasound, and laser ultrasound. These methods have slow detection efficiency and require a long time to occupy the track. Due to the pulse signal, there are certain detection blind spots. The magnetic particle testing method detects by forming a strong magnetic field near the steel rail and displays the medium. The detection instrument mechanism is relatively simple, the detection efficiency is low, and the magnetic field is easily affected by surrounding metal parts. The detection accuracy is low, the error is large, and the probability of using it in some important scenarios is low. The eddy current testing method distinguishes surface damage and cracks on steel rails based on the magnitude and phase of induced currents. This method has improved the detection efficiency to a certain extent. For example, the manually pushed rail inspection vehicle developed by the Sperry Corporation in the United States is based on eddy current testing technology [12], and Thomas et al., in Germany, ref. [13] have also conducted practical verification for contact fatigue cracks. Due to their simple structure and high detection efficiency, they are more susceptible to interference from the surrounding environment and have lower detection stability. The leakage magnetic detection method magnetizes the steel rail and determines whether there is crack damage by detecting the magnetic signal. This method has improved the detection efficiency to some extent. For example, the Sperry Corporation in the United States has developed a rail defect detection system by integrating ultrasonic detection with magnetic leakage detection [14]. Chen, W.C et al. [15] proposed a magnetization system to study the detection depth of transverse cracks in rails. However, this method still has the problem of low detection efficiency and poor detection stability. The radiographic testing method uses the ability of radiation to penetrate the steel rail to determine surface crack damage. This method has a simple structure and can also meet the requirements of faster detection. However, it has high requirements for installation equipment and poor stability. The track circuit [16] is used to detect whether the track is occupied and can also monitor the breakage of the rail online. However, this method has poor detection accuracy and cannot directly detect small cracks and damages on the surface of the rail. Acoustic emission technology compares and analyzes the collected elastic wave signals of steel rails through the emitted sound wave signals and then determines rail crack damage. This method is highly susceptible to external environmental interference, making it difficult to filter out noise during the detection process. Ultrasonic-guided wave detection technology determines the surface crack damage of steel rails by emitting ultrasonic waves propagating in waveguide media and receiving whether there are reflected waves. This method can detect the location of steel rail crack damage, but the detection efficiency is still low, and the stability is poor.

Non-contact detection methods mainly include image vision detection, which is widely used in non-destructive testing due to its advantages of fast detection speed and high detection accuracy [17]. For example, the rail inspection car developed by Chen, W.C [15] collects two-dimensional images of the rail through a camera, achieving automatic detection of surface defects on the rail, similar to the VIS comprehensive inspection system developed by ENSCO in the United States [18]. Hao, Q et al. [19] proposed a machine vision-based discrete surface defect detection system, Pathak, M et al. from Beijing Jiaotong University [20] designed a machine vision-based rail surface defect detection system, Tang, C et al. from Lanzhou Jiaotong University [21] proposed a rail surface defect detection method based on image grayscale gradient features, and Ye, J. et al. [22] proposed the detection of track damage based on 3D laser technology proposed a rail surface defect detection technology based on 2D/3D composite machine vision. Although image detection methods are widely used, they are mostly used for identifying and detecting surface cracks on rails. The specific location of cracks on the entire rail still needs to be studied, and the identification and detection of cracks with a width of 0.1 mm need to be addressed. 

Rail inspection counts a great deal for railway transportation, but several traditional and contact measurement techniques are still applied for rail damage monitoring, resulting in severe misjudgment, omission, and low speed. The existing technologies are facing lots of challenging tasks, including early detection, reliability, and system cost.

Many experts and scholars have carried out a great deal of research work in the field of rail damage defect detection and have also achieved certain research results. Yunpu Wu et al. [23] proposed a novel few-shot learning method for fault diagnosis of high-speed trains, which improves the accuracy and efficiency of fault diagnosis of high-speed trains. Fengyuan Zuo et al. [24] proposed a reliable method for detecting the key points of bolts in industrial magnetic separators, along with an intelligent monitoring model for robust detection of key bolt points. Different from previous methods that connect multiple tasks, such as introducing effective regularization techniques and a lightweight network structure in image processing, object detection, and key point extraction, this method improves the reliability of detection. Fengyuan Zuo et al. [25] proposed a multi-expert detection method based on X-rays for the automatic evaluation of welding defects in intelligent pipeline systems, which helps to improve the efficiency and accuracy of pipeline welding fault diagnosis. However, in the current daily maintenance and repair work of high-speed railways, manual maintenance and repair are still mainly relied on, and the automatic detection and repair of rail damage defects by robots have not been fully and truly realized. Therefore, the design and development of a robot for active rail damage detection is an important trend in future development.

In the above research methods, most of them are used to identify and detect surface damage of steel rails under static conditions and have been applied in some inspections. However, for the health monitoring and operation guarantee of railways, the most scientific and effective method is to dynamically identify and detect surface crack damage of steel rails in real-time online. Some researchers have conducted relevant research and achieved some research results, such as Xiao, J [26], using eddy contemporary thermal imaging technology to detect artificial cracks in steel rail samples at different excitation currents of 40 mm/s and 80 mm/s. Kim, S et al. [27] attempted to dynamically measure the induction thermography measurement system, while Lee, G et al. [28] studied the temperature distribution near defects in the dynamic detection mode of eddy contemporary thermography. This research also indicates that there is relatively little research on dynamic real-time identification and detection of rail surface crack damage, and there is even less research on the width, depth, size, and location of rail surface crack damage, especially in the fine identification and detection of rail surface crack damage.

The structure of this article is as follows. The second part introduces the structural design of the HRT robot body, driving mode, and control system. The third part analyzes the methods for detecting rail damage, including damage classification and changes in robot motion posture caused by different types of damage. In the fourth part, the rail damage detection capability and detection method of the proposed HRT robot were verified through experiments. Finally, the Conclusions section is presented.

## 2. Design of the HRT Robot

### 2.1. The Scheme of the HRT Robot

In the process of high-speed railway rail inspection, on the one hand, the system is required to be competent to run very stably and rapidly on the track and be competent to stop and start immediately. On the other hand, it is required to be able to accurately detect damage faults, especially to speedily and accurately detect damage faults in locations such as the rail surface. Therefore, the overall structure of the HRT robot adopts four independent track wheels, two of which are driving wheels, and the front “U”-shaped structure is equipped with a collection sensor and an LED strip light source.

The designed track inspection robot includes a main structure module, a drive module, a sensor module, and a light source control module. Figure 1 shows the physical prototype of the HRT robot.

The developed HRT robot realizes the detection of rail damage based on a multi-source data-driven method under the fusion of multi-source signals such as 2D optical signals, 3D laser signals, spectral signals, and vibration signals. The main structure of the HRT robot is built with 40 mm × 40 mm aluminum alloy material. The bottom plate, middle partition, and frame shell are all made of 3 mm 6061 aluminum alloy plate. The 6061 aluminum alloy plate has high hardness, strength, and a certain oxidation resistance. The bottom plate is reserved for mounting holes for motors, light source controllers, power-supplied controllers, control boards, etc., and industrial camera mounting brackets with slots are installed on the left and right side columns. The drive module includes two 24 V, 120 W DC motors, which are powered by a 24 V/30 A switching DC power supply, when the power supply is normal, the robot’s endurance time is 7 to 8 h. The motor is connected to the track wheel through an elastic coupling. The movement speed of the HRT robot is controlled by a speed regulator; under ideal conditions, the maximum speed calculated based on the theoretical design can reach 20 m/s. Detection at full speed will greatly improve the detection efficiency. The sensor module includes three XYZ three-axis acceleration sensors; the sensor is an XYZ three-axis piezoelectric acceleration sensor, model 1 A339 E. It has a sensitivity of 5 mV/m^2^ and can collect signals within a frequency range of 2–10,000 Hz. The feathery source module includes two LED strip white feathery sources and one 4-channel 0–255 controller, and the light source installation positions are designed to be adjustable, and the range of adjustment in the rail direction is 0–20 mm. The specific structural parameters of the HRT robot are shown in Table 1.

### 2.2. Sensors and Control Systems

In the design process of the HRT robot sensor module and control system, the first thing to consider is the start and stop and acceleration and deceleration of the HRT robot through wired and wireless remote control, as well as the real-time collection and processing of rail faults and damage and signal transmission feedback. Figure 2 shows the schematic diagram of the control system of the HRT robot.

The control system schematic diagram of the HRT robot is shown in Figure 2. The mega2560 board controls the movement of the HRT robot by wirelessly connecting through WiFi signals and controlling the forward and reverse rotation of the motor through PWM signals. By combining the XYZ three-axis acceleration sensor signal, the directions of the XYZ three-axis acceleration sensor are as follows: the forward direction of the testing robot is the X direction, the direction perpendicular to the steel rail is the Y direction, and the vertically upward direction is the Z direction. The real-time motion posture of the HRT robot is estimated to achieve online real-time detection of rail faults and damage.

## 3. Analysis of Rail Surface Damage Detection Methods

Rail damage defects include cracks, wear, geometric irregularities, crushing, depressions (or dents), wave wear, bending deformation, surface defects, tilt, misalignment, gaps, external corrosion, and other damages. These damage defects will affect the posture of the inspection robot. This paper proposes a track defect detection method based on robot posture. First, the possible defects of the rails are classified; then, the robot posture changes, caused by each defect, are modeled; finally, the relationship between the robot posture change and the defect geometric parameter is established.

### 3.1. Classification of Rail Defects and Damage

The surface damage of rails can be summarized into six fundamental types, namely, surface protrusion damage (SD), surface depression damage or depression damage, crush defect damage (CD), surface gap defect damage (ID), bending defect damage (BD), surface cracked damage (CRD), and surface dislocation damage (DD). As shown in Table 2, these are the six fundamental types of defects, among which surface crack defects include parallel crack defects (PCDs; the crack is in the same direction as the rail), vertical crack defects (VCDs; the crack direction is perpendicular to the rail direction), and acute-angle crack defects (ACDs; the angle between the crack and the rail direction is an acute angle). Surface bending defect types included left bending defect damage (LBD), right bending defect damage (RBD), further-up bending defect damage (UBD), and subordinate bending defect damage (DBD). Dislocation defect damage types include further-up dislocation defect damage (UDD) and subordinate dislocation defect damage (DDD). This paper will focus on analyzing the motion state and motion force of the HRT robot during its operation on the surface of the above-mentioned defect-damaged rails.

### 3.2. Kinematic Analysis of HRT Robots

Through specific analysis of the above-mentioned defects in the steel rail, for HRT robots operating on the track, each type of defect in the steel rail is a relative obstacle to the HRT robot during its operation on the track. The obstacles during the HRT robot’s operation on the track are classified into five types of obstacles, which are the five obstacle-crossing motion states. They are, respectively, hill climbing obstacle motion (CS), upper step obstacle motion (US), subordinate step obstacle motion (DS), groove gap obstacle motion (TC), and subordinate slope obstacle motion (DO). In the analysis of the HRT robot’s motion state and posture changes when encountering various defect obstacles during its movement on the track, the main focus is on analyzing the changes in its motion state and posture.

For hill climbing obstacle motion (CS), as shown in Figure 3, the transient posture diagram of the HRT robot with a hill climbing obstacle motion time in moments is shown. L is the front and rear wheelbase of the HRT robot, and the upward bending angle of the steel rail is γ. The angle between the HRT robot and the steel rail is θ, $h_{1}$ is the height difference between the front and rear axles of the HRT robot, and the acceleration during climbing obstacles is a; the speed is ν. The HRT robot’s acceleration a and speed ν can be expressed as follows:(1)$$\left\{\begin{array}{l} h_{1}=S\cdot \sin\gamma \\ S=\frac{1}{2}at^{2}\\ h_{1}=L\cdot \sin\theta \\ \nu =a\cdot t\\ a=\frac{2L\cdot \sin\theta }{t^{2}\cdot \sin\gamma }\\ \nu =\frac{2L\cdot \sin\theta }{t\cdot \sin\gamma } \end{array}\right.$$

For the step-up obstacle motion (US), as shown in Figure 4, the transient posture diagram of the HRT robot with a step-up obstacle motion time of t moments is shown. L represents the front and rear wheelbase of the HRT robot, and the height of the step where the steel rail forms a “step” defect is h; the height difference between the front and rear axles of the HRT robot after completing the step-up motion is $h_{1}$; $\alpha_{1}$ is the angle between the wheel radius and the steel rail at the point where the robot just touches the step; $\alpha_{2}$ is the difference between the radius of the rear wheel and the angle of the steel rail for the robot to complete the step-up movement and the radius of the front wheel and the angle of the steel rail for the step-up movement. The angle between the HRT robot and the steel rail after completing the step-up obstacle movement is θ, the HRT robot’s wheel radius is r, the acceleration during the upper stage motion is a, and the speed is ν. The HRT robot’s acceleration a and speed ν can be expressed as follows:(2)$$\left\{\begin{array}{l} h_{1}=L\cdot \sin\theta \\ h=r\cdot \sin\left(\alpha_{1}+\alpha_{2}\right)-r\cdot \sin\alpha_{1}\\ \sin\alpha_{1}=\frac{r-h}{r}\\ \cos\alpha_{2}=\frac{r^{2}+r^{2}-S^{2}}{2\cdot r\cdot r}\\ S=\frac{1}{2}\cdot at^{2}\\ v=at\\ a=\frac{2\sqrt{2}\cdot r\cdot \sqrt{1-\cos\alpha_{2}}}{t^{2}}\\ \nu =\frac{2\sqrt{2}\cdot r\cdot \sqrt{1-\cos\alpha_{2}}}{t} \end{array}\right.$$

For the descending step obstacle movement (DS), as shown in Figure 5, the transient posture diagram of the HRT robot’s descending step obstacle movement at time t is shown. L represents the front and rear wheelbase of the HRT robot, and the height of the step where the steel rail forms a “step” defect is h; the height difference between the front and rear axles of the HRT robot after completing the down step movement is $h_{1}$; α is the angle between the center of the front and rear wheels when the robot completes the downward step movement; $\alpha_{1}$ is the angle between the robot and the steel rail after completing the down step movement, and the angle between the HRT robot and the steel rail after completing the down step obstacle movement is θ; the HRT robot’s wheel radius is r; the acceleration during the upper stage motion is a, and the speed is ν; the HRT robot’s acceleration a and speed ν can be expressed as follows:(3)$$\left\{\begin{array}{l} \sin\theta =\frac{h_{1}}{L}\\ h_{1}=r-r\cdot \cos\alpha \\ \sin\alpha_{1}=\frac{r-h}{r}\\ S=\cos\alpha_{1}\cdot r\\ \cos\alpha_{1}=\frac{\sqrt{r^{2}-{\left(r-h\right)}^{2}}}{r}\\ S=\frac{1}{2}\cdot at^{2}\\ v=at\\ a=\frac{2\sqrt{r^{2}-{\left(r-h\right)}^{2}}}{t^{2}}\\ v=\frac{2\sqrt{r^{2}-{\left(r-h\right)}^{2}}}{t} \end{array}\right.$$

For the groove gap obstacle motion (TC), as shown in Figure 6, the transient posture diagram of the HRT robot with a groove gap obstacle motion time of t is shown. L represents the front and rear wheelbase of the HRT robot, and the maximum height at which the HRT robot descends during the process of crossing the groove gap is h; the distance between gaps is d; $\alpha_{1}$ is the angle between the robot and the steel rail at the maximum height of descent during the process of crossing the groove gap; the angle between the maximum descending height of the HRT robot and the steel rail during the process of crossing the groove gap is θ; the HRT robot’s wheel radius is r; the acceleration during the upper stage motion is a; the speed is ν. The HRT robot’s acceleration a and speed ν can be expressed as follows [29]:(4)$$\left\{\begin{array}{l} h=r-L\cdot \sin\theta \\ \cos\alpha_{1}=\frac{h}{r}\\ h=\frac{d}{2\tan\alpha_{1}}\\ S=\sqrt{2}\cdot r\cdot \sqrt{1-\cos\alpha_{1}}\\ S=\frac{1}{2}\cdot at^{2}\\ v=at\\ a=\frac{2\sqrt{2}\cdot r\cdot \sqrt{1-\cos\alpha_{1}}}{t^{2}}\\ v=\frac{2\sqrt{2}\cdot r\cdot \sqrt{1-\cos\alpha_{1}}}{t} \end{array}\right.$$

For the downward slope obstacle motion (DO), as shown in Figure 7, the transient posture diagram of the HRT robot’s downward slope obstacle motion time is t. L represents the front and rear wheelbase of the HRT robot, and the downward bending angle of the steel rail forms a slope of $\alpha_{1}$; the angle between the HRT robot and the steel rail is θ; h is the height difference between the front and rear axles of the HRT robot, and the acceleration during climbing obstacles is a; the speed is ν. The HRT robot’s acceleration a and speed ν can be expressed as follows [30]:(5)$$\left\{\begin{array}{l} h=L\cdot \sin\theta \\ \sin\alpha_{1}=\frac{h}{S}\\ S=\frac{1}{2}\cdot at^{2}\\ v=at\\ a=\frac{2\cdot L\cdot \sin\theta }{t^{2}\cdot \sin\alpha_{1}}\\ v=\frac{2\cdot L\cdot \sin\theta }{t\cdot \sin\alpha_{1}} \end{array}\right.$$

### 3.3. Analysis of the Relationship Between the Motion Posture of HRT Robots and Rail Defects

As shown in Table 1, a total of 12 types of surface damage to high-speed train rails are listed, which are summarized under different working conditions. According to the motion postures of 12 types of damages encountered by HRT robots during their operation on the track, they are divided into 5 categories: climbing obstacle motion (CS), upper step obstacle motion (US), lower step obstacle motion (DS), groove gap obstacle motion (TC), and downward slope obstacle motion (DO). These 12 types of rail surface damage are found in over 98% of high-speed train rail damage, so we specifically analyze and study these 5 types of rail damage, focusing on the changes in the movement posture of HRT robots when encountering various types of damage and analyzing and developing the relationship between the movement posture of HRT robots and rail damage.

The first type of rail defect is hill climbing obstacle motion (CS), which means that the rail forms a certain angle with the horizontal plane γ. The motion attitude parameters of HRT robots passing through track defects can be calculated by Formula (1). As shown in Figure 8, the HRT robot is at an angle of γ. The steel rail forms a certain angle with the horizontal plane, with an initial value of 0. As the robot moves forward, the acceleration value increases continuously; $a_{0.05{}^{\circ}}\max=0.06\;{m/s}^{2}$, $a_{0.1{}^{\circ}}\max=0.12\;{m/s}^{2}$, and $a_{0.15{}^{\circ}}\max=0.26\;{m/s}^{2}$; the speed value also increases continuously and continues to decrease after reaching its peak; $\nu_{0.05{}^{\circ}}\max=0.04\;m/s,\;\nu_{0.1{}^{\circ}}\max=0.06\;m/s$, and $\nu_{0.15{}^{\circ}}\max=0.12\;m/s$. From the curve, it can be seen that the acceleration and velocity motion curves of the robot undergo significant abrupt changes at 4.74 s, indicating that it is passing through the CS defect at 4.74 s and finally approaching the initial value state, as shown in Figure 9.

The second type of rail defect is the upward step obstacle movement (US), which means that there is a certain height difference between the rail and the horizontal plane h. The motion attitude parameters of HRT robots passing through track defects can be calculated by Formula (2). As shown in Figure 10, the HRT robot has a height difference of h, which is 0.5 mm, 1.0 mm, or 1.5 mm, and the initial value of the height difference between the steel rail and the horizontal plane is 0. As the robot moves forward, the acceleration of the robot will undergo a sudden change when encountering step obstacles, and then the acceleration value will continue to increase in a short period of time; $a_{0.5}\max=0.12\;{m/s}^{2}$, $a_{1.0}\max=0.26\;{m/s}^{2}$, and $a_{1.5}\max=0.24\;{m/s}^{2}$. After reaching its peak, i.e., after passing through the step obstacle, the acceleration returns to its initial value. The speed value also increases continuously, reaching its peak and continuing to decrease to the initial value, $\nu_{0.5}\max=0.03\;m/s$, $\nu_{1.0}\max=0.1\;m/s$, or $\nu_{1.5}\max=0.05\;m/s$, until reaching the initial value state at the end. From the curve, it can be seen that the acceleration and velocity motion curves of the robot undergo significant abrupt changes at 4.638 s, indicating that it is passing through the CS defect at 4.638 s and finally approaching the initial value state, as shown in Figure 11.

The third type of rail defect is the descending step obstacle movement (DS), which means that there is a certain height difference between the rail and the horizontal plane h; the motion attitude parameters of HRT robots passing through track defects can be calculated by Formula (3). As shown in Figure 12, the HRT robot has a height difference of h, which is 0.5 mm, 1.0 mm, or 1.5 mm; the initial value of the height difference between the steel rail and the horizontal plane is 0. As the robot moves forward, the acceleration of the robot will undergo a sudden change when encountering step obstacles, and then the acceleration value will continue to increase in a short period of time; $a_{0.5}\max=0.10\;{m/s}^{2}$, $a_{1.0}\max=0.28\;{m/s}^{2}$, and $a_{1.5}\max=0.32\;{m/s}^{2}$; after reaching its peak and after passing through the step obstacle, the acceleration returns to its initial value. The speed value also increases continuously, reaching its peak and continuing to decrease to the initial value, $\nu_{0.5}\max=0.06\;m/s$, $\nu_{1.0}\max=0.11\;m/s$, or $\nu_{1.5}\max=0.03\;m/s$, until reaching the initial value state at the end. From the curve, it can be seen that the acceleration and velocity motion curves of the robot undergo significant sudden changes at 5.77 s, indicating that it is passing through the DS defect at 5.77 s and finally approaching the initial value state, as shown in Figure 13.

The fourth type of rail defect is groove gap obstacle motion (TC), which means that there is a certain distance d of groove gaps on the surface of the rail. The motion attitude parameters of the HRT robot when passing through the rail defect can be calculated by a certain formula. As shown in Figure 14, the HRT robot operates at a gap distance d, which is 0.5 mm, 1.5 mm, or 2.5 mm; the motion posture parameters of the robot will undergo a sudden change in acceleration when encountering obstacles in groove gaps as the robot’s forward distance increases, and then the acceleration value will continue to increase in a short period of time; $a_{0.5}\max=0.09\;{m/s}^{2}$, $a_{1.5}\max=0.21\;{m/s}^{2}$, $a_{2.5}\max=0.26\;{m/s}^{2}$; after reaching the peak value, that is, after passing through the groove gap obstacle, the acceleration returns to the initial value. The speed value also increases continuously, reaching its peak and continuing to decrease to the initial value, $\nu_{0.5}\max=0.03\;m/s$, $\nu_{1.0}\max=0.08\;m/s$, or $\nu_{1.5}\max=0.04\;m/s$, until reaching the initial value state at the end. From the curve, it can be seen that the acceleration and velocity motion curves of the robot undergo significant abrupt changes at 7.253 s, indicating that it is passing through the TC defect at 7.253 s and finally approaching the initial value state, as shown in Figure 15.

The fifth type of rail defect is the downward slope obstacle movement (DO), which means that the rail forms a certain angle with the horizontal plane γ. The motion attitude parameters of HRT robots passing through track defects can be calculated by Formula (5). As shown in Figure 16, the HRT robot is at the included angle γ, which is 0.05°, 0.1°, or 0.15°. The initial value of the angle formed between the steel rail and the horizontal plane is 0. As the robot moves forward, the acceleration of the robot will undergo a sudden change when encountering downhill obstacles, and then the acceleration value will continue to increase in a short period of time; $a_{0.05{}^{\circ}}\max=0.09\;{m/s}^{2}$, $a_{0.1{}^{\circ}}\max=0.19\;{m/s}^{2}$, and $a_{0.15{}^{\circ}}\max=0.14\;{m/s}^{2}$; the speed value also increases continuously and continues to decrease after reaching its peak, $\nu_{0.05{}^{\circ}}\max=0.05\;m/s$, $\nu_{0.1{}^{\circ}}\max=0.09\;m/s$, or $\nu_{0.15{}^{\circ}}\max=0.10\;m/s$. From the curve, it can be seen that the acceleration and velocity motion curves of the robot undergo significant abrupt changes at 5.753 s. It can be determined that the robot is passing through the DO defect at 5.753 s and finally approaching the initial value state, as shown in Figure 17.

## 4. The Experiments to Verify the RIR Robot and Detection Method

In order to test and verify the motion posture and rail fault detection method of the HRT robot on the track, a testing test platform was built. The steel rail was fixed on the ground, with a total length of 3 m, and composed of 4 sections, 2 sections of 2 m and 2 sections of 1 m, which can be used to construct rail groove gap defects, step defects, and slope defects for the experimental verification of HRT robot testing. The test experimental platform in Figure 18a–h shows the motion posture of the HRT robot running on the track. The average speed of the HRT robot during operation is 0.08 m/s. During the experiment, a manual power control line was specially set up to cut off the power in real-time in case of emergency. Within 0–3 s, the HRT robot operates normally on the track. Within 3–9 s, the robot is about to pass through various rail faults and defects. Within 9–5 s, the HRT robot will complete various rail fault and defect obstacles. Within 15–18 s, after successfully completing various rail faults and defects, the HRT robot operates normally. Throughout the entire testing process, the motion stability of the HRT robot is not affected by rail faults and defects. However, various rail faults and defects can affect the motion posture of the HRT robot, mainly including sudden changes in acceleration and speed when passing through a fault and defect.

During the HRT robot experimental testing process, five types of fault defects, TC, CS, US, DS, and DO, were set on a 3 m steel rail. The HRT robot was tested twice, and the motion acceleration and velocity changes of the HRT robot after passing through the five types of fault defects were recorded and statistically analyzed. The motion acceleration analysis results are shown in Figure 19, and the motion velocity analysis results are shown in Figure 20. When the HRT robot went through five types of fault defects, the acceleration suddenly changed, exceeding the ideal value, and an error rate of 97% was achieved in two experiments with the same fault defects. When the HRT robot underwent five types of fault defects, its speed underwent sudden changes, exceeding or approaching the ideal value, and the error rate of the same fault defect in two experiments was 98%. In Figure 19 and Figure 20, the horizontal axis represents the displacement distance of the robot. The physical quantity of the displacement distance is denoted by “s”, and the unit is meters (m).

## 5. Conclusions

A new type of rail detection robot is proposed to address the issue of surface damage of the rail head detection in high-speed railway systems, and a fault detection method that utilizes the robot’s own posture changes to evaluate rail defects is proposed. The mechanical structure of the HRT robot and the control system of the HRT robot were designed. The types of surface damage of the rail head in high-speed railway systems were classified, and the motion characteristics of HRT robots passing through track defects were analyzed. We analyzed the variation curves of attitude acceleration and velocity of robots passing through different track defects. Finally, the detection ability of HRT robots in track defect detection methods was verified through experiments; when the HRT robot passed through five types of fault defects, its error rate was within the range of 2–3%, meeting the design expectations. HRT robots and corresponding track fault detection methods cannot only be applied to track fault detection in high-speed railway systems but also to track fault detection in conventional trains, railways, bridges, etc. The 12 types of rail damage defects and the five major categories of damage defects summarized and classified are mainly analyzed according to the requirements in the IRS 70712 [31] and EN 17397-1:2020 [30] standards, including the damage defects that need to be detected. Therefore, the proposed method is effective for the detection of the existing surface damage defects of high-speed railway rails.

Due to the limitations of the experimental verification test environment, although the verification of the method proposed in this article has been completed through experimental tests, there is still a lot of room for improvement. The focus and direction of future research will be on how to use new methods under multi-source data fusion to identify and detect rail damages, how to propose new algorithms for rail damage identification, and how to develop a new type of intelligent rail inspection, maintenance, and operation robot with a higher detection speed, higher identification and detection accuracy, and higher identification accuracy on actual railway lines.

## Figures and Tables

**Figure 1 sensors-25-03063-f001:**
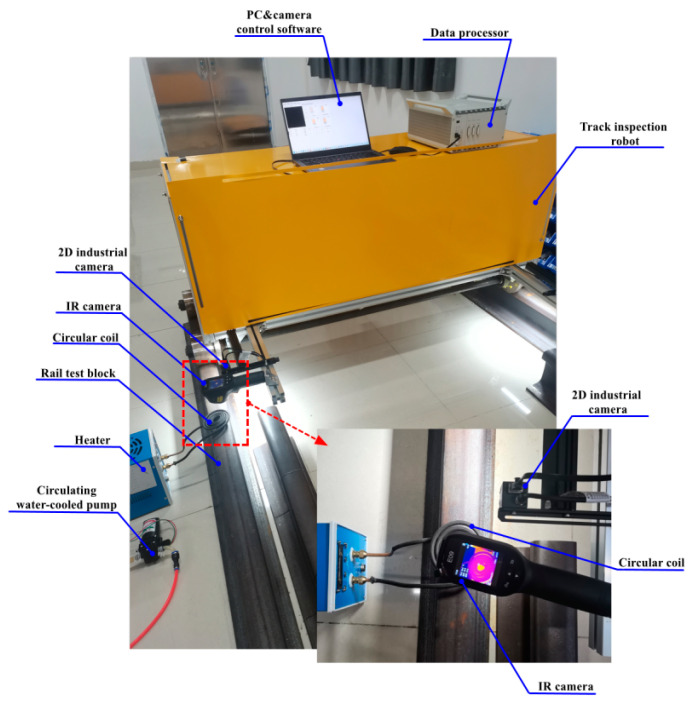
The prototype of the HRT robot.

**Figure 2 sensors-25-03063-f002:**
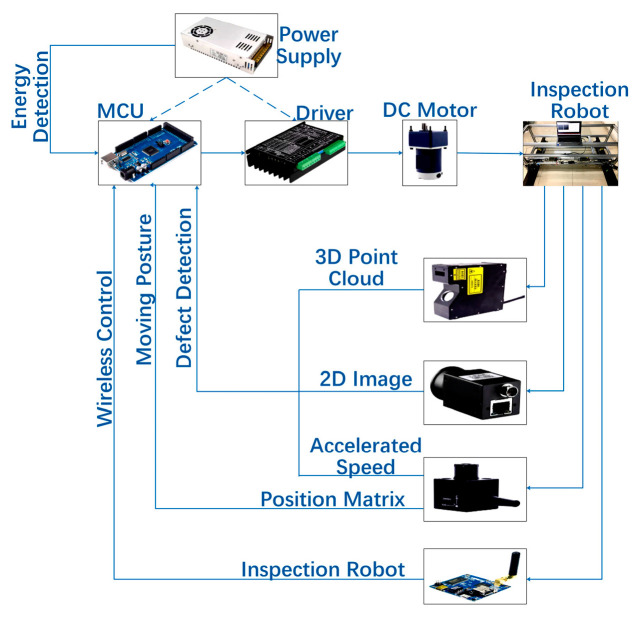
Control system schematic diagram of HRT robot.

**Figure 3 sensors-25-03063-f003:**
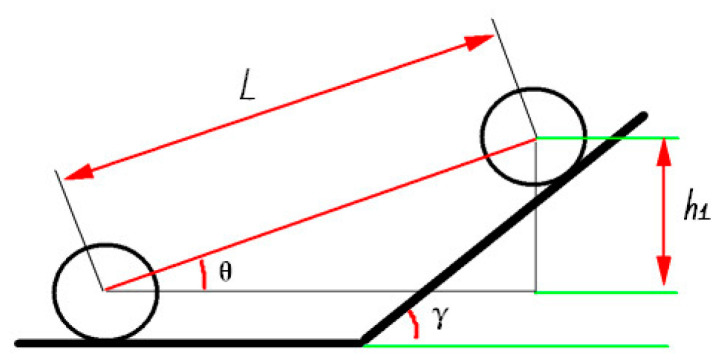
Schematic diagram of climbing obstacle movement (CS).

**Figure 4 sensors-25-03063-f004:**
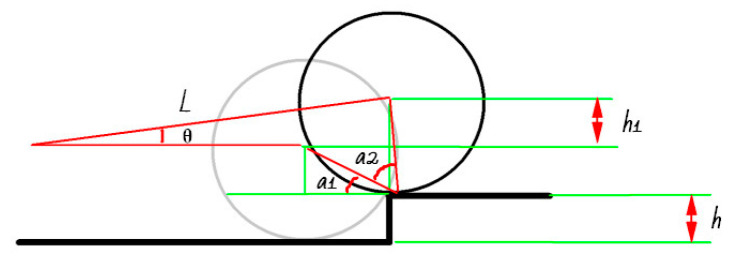
Schematic diagram of obstacle movement on stairs (US).

**Figure 5 sensors-25-03063-f005:**
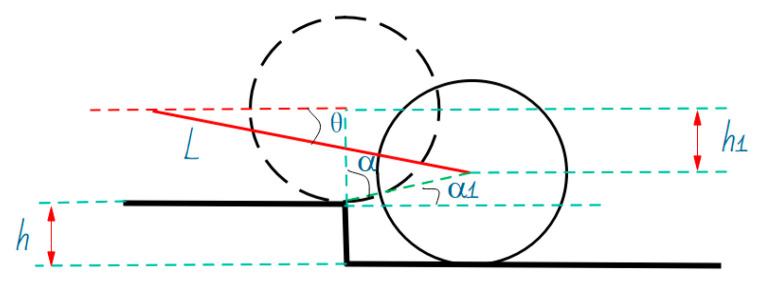
Schematic diagram of obstacle movement in the lower steps (DS).

**Figure 6 sensors-25-03063-f006:**
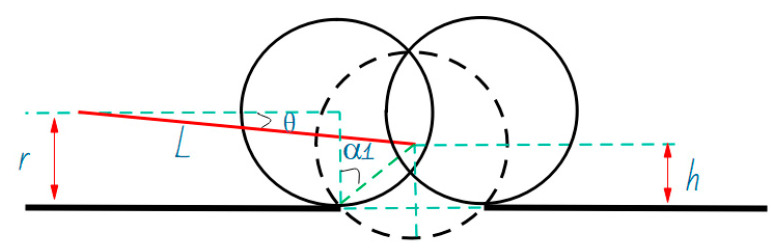
Trough gap obstacle movement (TC).

**Figure 7 sensors-25-03063-f007:**
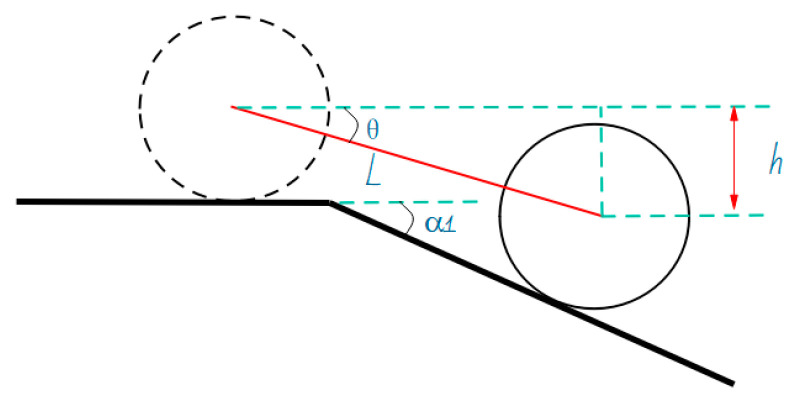
Downhill obstacle movement (DO).

**Figure 8 sensors-25-03063-f008:**
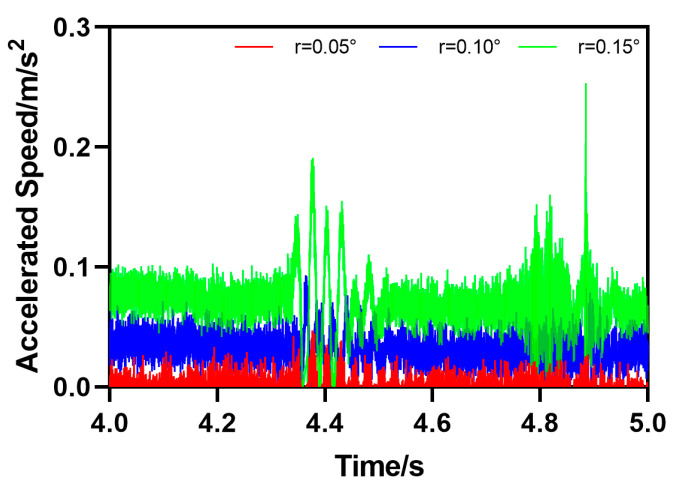
Acceleration parameter curve of HRT robot during climbing obstacle motion (CS).

**Figure 9 sensors-25-03063-f009:**
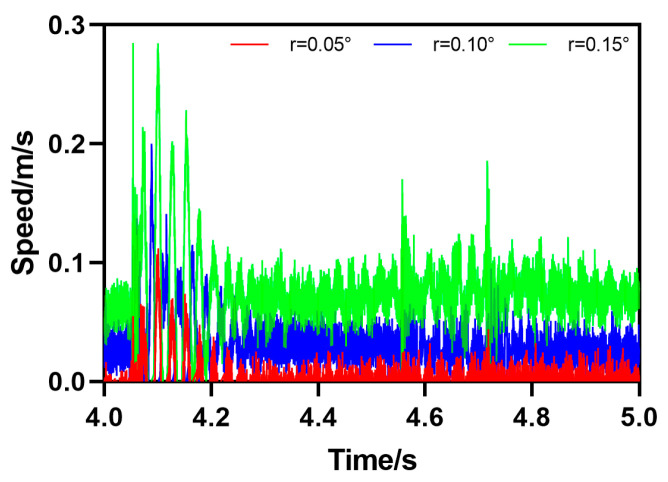
Speed parameter curve of HRT robot moving through climbing obstacles (CS).

**Figure 10 sensors-25-03063-f010:**
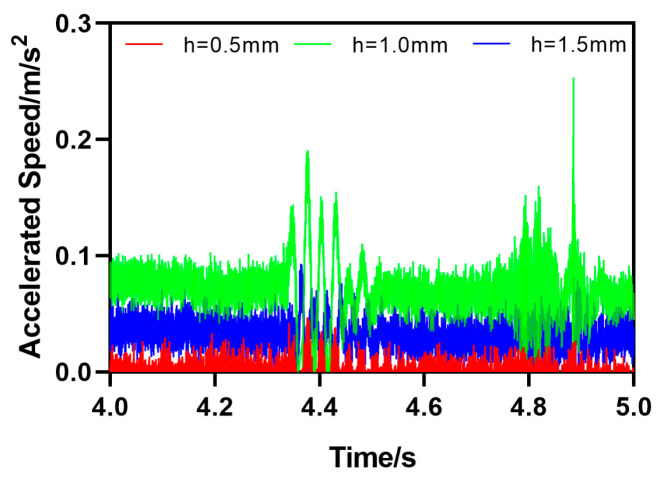
Acceleration parameter curve of HRT robot moving through an obstacle on an upward step (US).

**Figure 11 sensors-25-03063-f011:**
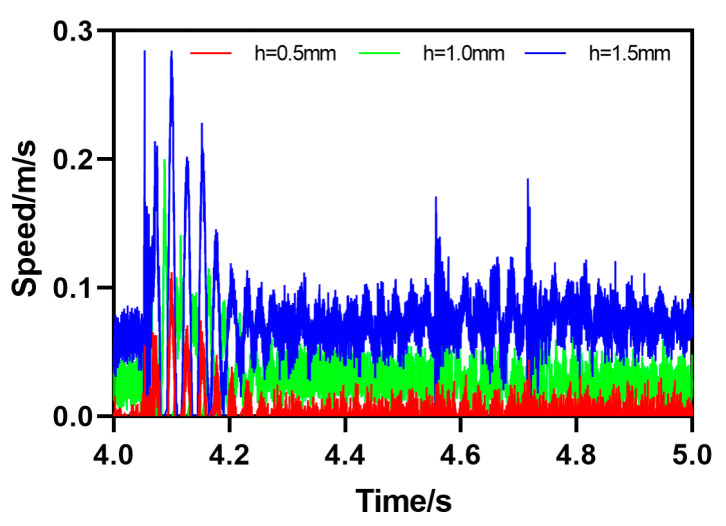
Speed parameter curve of HRT robot passing through an obstacle on an upward step (US).

**Figure 12 sensors-25-03063-f012:**
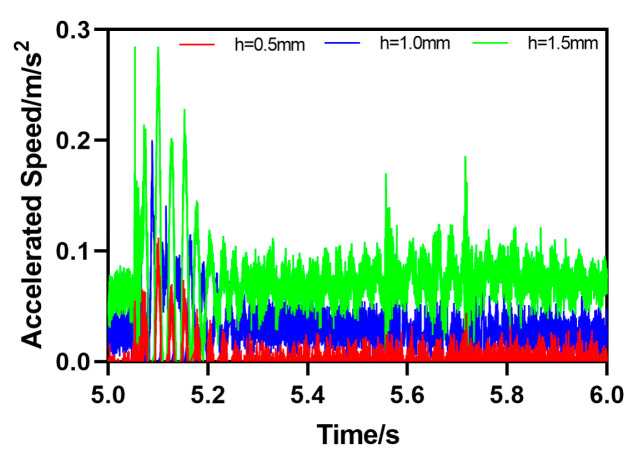
Acceleration parameter curve of HRT robot moving through a lower step obstacle (DS).

**Figure 13 sensors-25-03063-f013:**
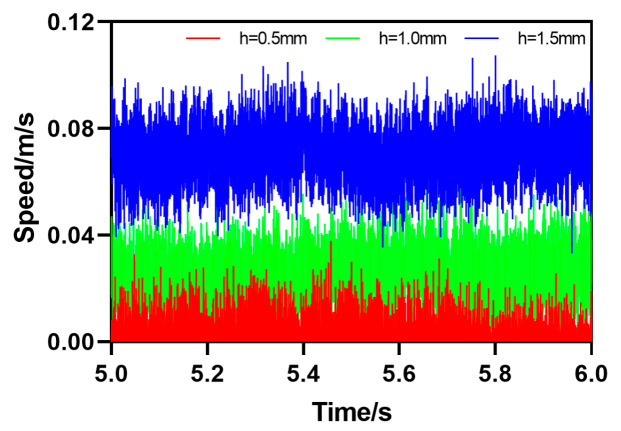
Speed parameter curve of HRT robot passing through a downward step obstacle motion (DS).

**Figure 14 sensors-25-03063-f014:**
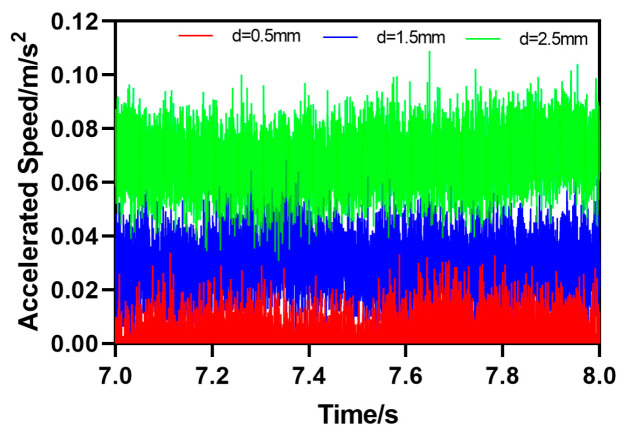
Acceleration parameter curve of HRT robot moving through groove gap obstacle (TC).

**Figure 15 sensors-25-03063-f015:**
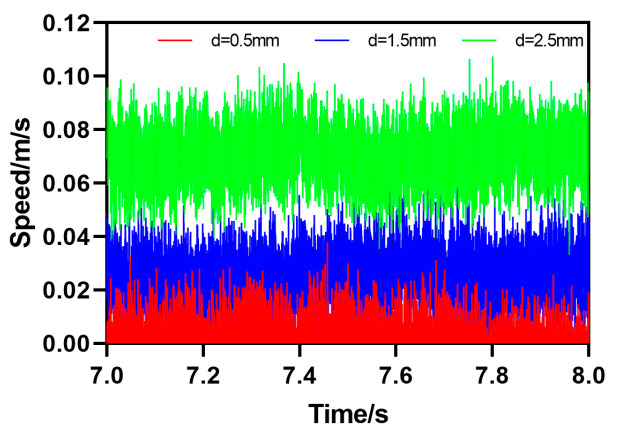
Speed parameter curve of HRT robot moving through groove gap obstacles (TC).

**Figure 16 sensors-25-03063-f016:**
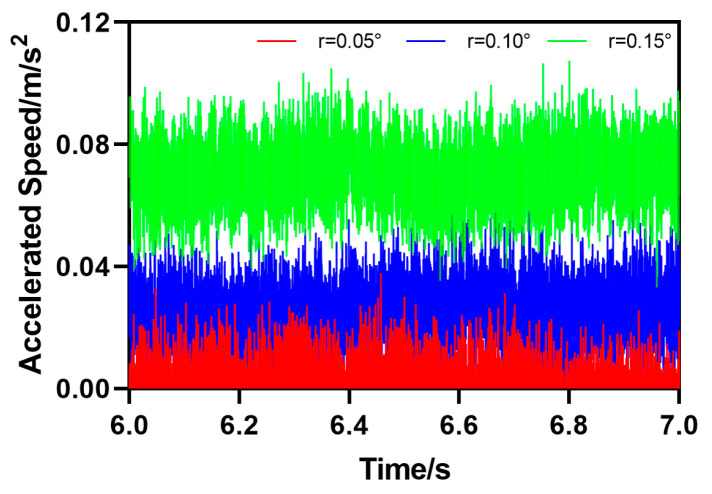
Acceleration parameter curve of HRT robot moving through downhill obstacles (DO).

**Figure 17 sensors-25-03063-f017:**
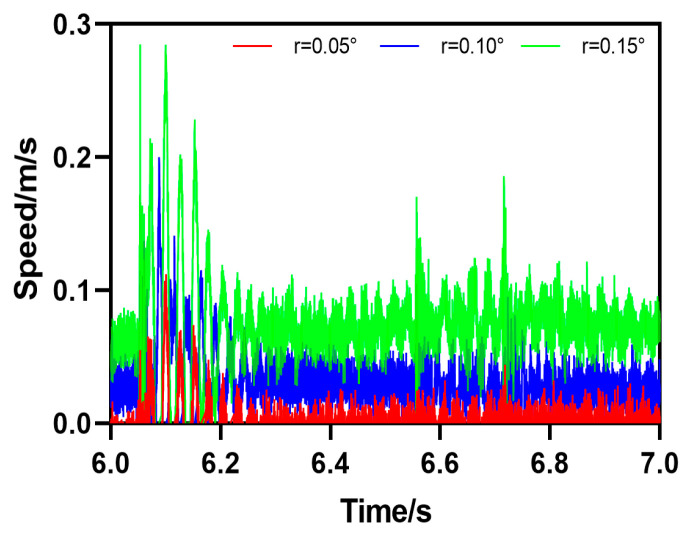
Speed parameter curve of HRT robot passing through downhill obstacle movement (DO).

**Figure 18 sensors-25-03063-f018:**
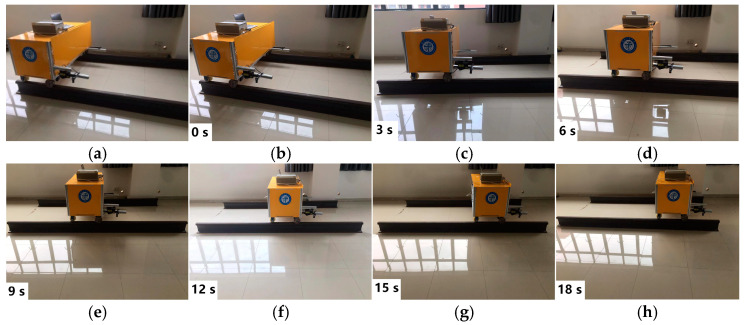
Process diagram of HRT robot crossing track defect obstacles (**a**) Starting point position; (**b**) 0 s position; (**c**) 3 s position; (**d**) 6 s position; (**e**) 9 s position; (**f**) 12 s position; (**g**) 15 s position; (**h**) 18 s position.

**Figure 19 sensors-25-03063-f019:**
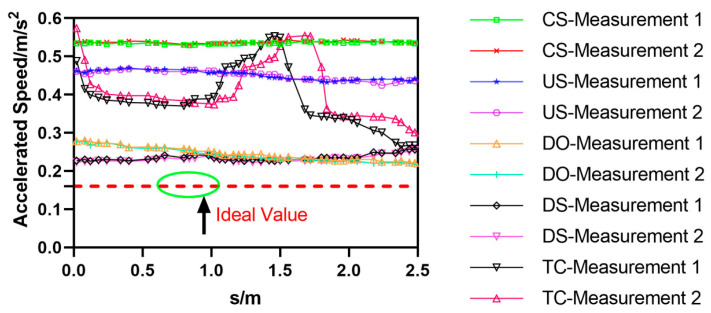
Acceleration and attitude curve of HRT robot crossing rail defect obstacle.

**Figure 20 sensors-25-03063-f020:**
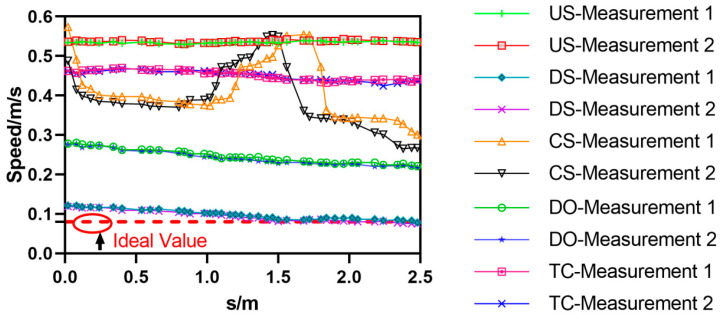
Speed and attitude curve of HRT robot when crossing track defect obstacles.

**Table 1 sensors-25-03063-t001:** HRT robot parameters.

Symbol	Parameters	Value
M (kg)	the mass of the HRT robot	30
LHRT (mm)	the length of the HRT robot	580
WHRT (mm)	the width of the HRT robot	1515
H HRT (mm)	the height of the HRT robot	670
L1 (mm)	the wheelbase	400
r (mm)	the radius of wheels	450
L2 (mm)	the track wheel center distance	1435

**Table 2 sensors-25-03063-t002:** Typical defects of rails.

SerialNumber	DefectName	Defect Type	Schematic Diagram
1	SD	Step defect	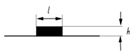
2	CD	Compression defect	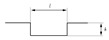
3	ID	Interstitial defect	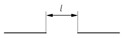
4	LBD	Left bending defect	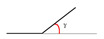
5	RBD	Right bending defect	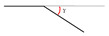
6	PCD	Parallel crack defect	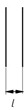
7	VCD	Vertical crack defect	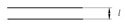
8	ACD	Acute Angle crack defect	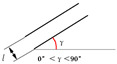
9	UDD	Upward dislocation defect	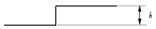
10	DDD	Downward dislocation defect	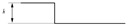
11	UBD	Upward bending defect	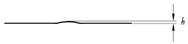
12	DBD	Downward bending defect	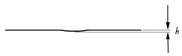

## Data Availability

Not applicable.

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
