# Peer review of "A Novel Rail Damage Fault Detection Method for High-Speed Railway"

_sensors, 2025, doi:10.3390/s25103063_

Round 1
Reviewer 1 Report (New Reviewer)
Comments and Suggestions for Authors
The manuscript is interesting and well-written. The article concerns a specialist robot for inspection of surface damage to the rail head. It is a significant problem in the quick diagnostics of track infrastructure and the detection of defects, especially for high-speed lines.
The submitted research and its results are an appropriate resource for further study. The following comments are suggested for authors for improve article
- The presented classification of rail defects should be related to the requirements of the IRS 70712 and EN 17397-1:2020 standards. According to the classification of the authors of the article, this method applies only to surface damage visible on the surface of the rail head.
- What experimental studies were performed and what measuring equipment was used (inspection railcar or other) that were compared with the proposed solution?
- Does Gap Obstacle Movement (figure 6) apply to the passage of a wheel through the joint between rails? On high-speed lines, continuous welded rail (CWR) is used to reduce the dynamic effect of the wheel passing through such a rail joint.
- There are no references in the text to bibliography items [5,6,8,9].
- Is the schematic diagram of obstacle movement on stairs, presented the robot entering the railway slope line elevation (figure 4 and 5), or other obstacle ?
Author Response
Comments 1: [The presented classification of rail defects should be related to the requirements of the IRS 70712 and EN 17397-1:2020 standards. According to the classification of the authors of the article, this method applies only to surface damage visible on the surface of the rail head.]
Response 1:[Thank you very much for the professional suggestions given by the reviewers.
The method proposed in the article is mainly used for the detection of surface damage defects of the rails. Certainly, based on the research in this article, further studies will be carried out in combination with the actual situation in the future. Developing methods for identifying and detecting both surface and internal damage of the rails will be the content to be further studied and overcome in the future.
Thank you very much for the extremely professional comments and suggestions from the reviewers! Thank you!]
Comments 2:[What experimental studies were performed and what measuring equipment was used (inspection railcar or other) that were compared with the proposed solution?]
Response 2:[Thank you very much for the professional suggestions given by the reviewers.
The 12 types of surface damage defects of the rails are classified into 5 major categories. They are respectively hill climbing obstacle motion (CS), upper step obstacle motion (US), subordinate step obstacle motion (DS), groove gap obstacle motion (TC), and subordinate slope obstacle motion (DO). Experimental tests and research have been carried out for each of the 5 major categories of damage defects. During the experiment, a vibration signal tester, the independently developed RIR Robot platform, and the X-Y-Z three-axis acceleration sensor were used for experimental testing and verification.
Thank you very much for the extremely professional comments and suggestions from the reviewers! Thank you!]
Comments 3:[Does Gap Obstacle Movement (figure 6) apply to the passage of a wheel through the joint between rails? On high-speed lines, continuous welded rail (CWR) is used to reduce the dynamic effect of the wheel passing through such a rail joint.]
Response 3:[Thank you very much for the professional suggestions provided by the reviewers.
The groove gap obstacle motion (Figure 6) is applicable not only to the detection of the wheel passing through the rail joint, but also to the detection of the gap defects caused by the actual damage of the weld seam. It is also applicable to the identification and detection of the gap defects generated by the surface cracks.On high-speed railway lines, continuously welded rails (CWR), also known as seamless rails, are used to reduce the dynamic effect when the wheels pass through such rail joints. When there are damage and gap defects in the weld seam, they can be identified and detected.
Thank you very much for the extremely professional comments and suggestions from the reviewers! Thank you!]
Comments 4:[There are no references in the text to bibliography items [5,6,8,9]
Response 4:[Thank you very much for the professional suggestions given by the reviewers.
Ruipeng Gao [5] and others analyzed the surface crack damage of the rail by studying the relationship between the fatigue crack propagation process of the rail and the number of cyclic loads. Xuefeng Ni [6] and others proposed a novel algorithm for detecting rail surface defects based on partitioned edge features (PEF) to solve the problem of identifying and detecting rail surface damage. Mariusz Nieniewski [8] and others developed a system that uses morphological operations to detect rail defects and extract their shapes, enabling the detection of surface damage defects on the rails. Xiating Jin [9] and others proposed a method of directly performing parameter learning in the Expectation-Maximization (EM) algorithm. Meanwhile, they employed the Faster Region-based Convolutional Neural Network (Faster RCNN) with a parallel structure to realize the target location of the damage on the rail surface, thereby enhancing the capability of detecting rail surface defects. The relevant content has been revised and supplemented in the article. Thank you very much for the extremely professional comments and suggestions from the reviewers! Thank you!]
Comments 5:[Is the schematic diagram of obstacle movement on stairs, presented the robot entering the railway slope line elevation (figure 4 and 5), or other obstacle ?]
Response 5:[Thank you very much for the professional suggestions provided by the reviewers.
Here it mainly refers to the damage defects such as pits or bulges that appear on the surface of the rails, which form such similar damage defects. This also includes situations such as the slopes on the rail surface caused by the unevenness of the railway line. Thank you very much for the extremely professional comments and suggestions from the reviewers! Thank you!]
Reviewer 2 Report (New Reviewer)
Comments and Suggestions for Authors
This paper proposes a rail inspection robot method for track geometry condition assessment, primarily focusing on detecting local geometric defects in rails. The study initially identifies 12 common types of rail head surface defects and categorizes them into five major classifications through feature analysis. It then conducts kinematic analysis of the robot's movement patterns under these five defect categories, investigating the correlation between characteristic defect types and robotic motion postures during detection. However, the paper fails to clearly demonstrate the research significance and academic value:
- Representativeness of defect classification: The representativeness of the five categorized rail head surface defects requires scrutiny. These defects (all causing abrupt motion changes in the HRT robot) belong to surface geometric anomalies that typically generate significant vibrations during train operations. Current maintenance practices already detect such defects using conventional track inspection vehicles. The paper lacks justification for the necessity of employing HRT robots for precise detection and classification of these defects.
- Curves and gradients parameter exclusion: The study neglects to address how to distinguish and eliminate the influence of widespread track features like curves and gradients in actual track lines from the identified five defect categories.
- High-speed railway relevance: No evident connection is established between the proposed defect classifications and their specific implications for high-speed railway operations.
- Robotic intelligence deficiency: The described HRT robot essentially functions as a conventional track inspection vehicle with mobility, without demonstrating intelligent features that would qualify it as a robotic system in terms of operational procedures or functionality.
- Academic depth limitation: The data analysis methods presented for the five defect types appear superficial, lacking sophisticated analytical approaches that would enhance academic rigor.
In conclusion, the proposed novel track inspection robot fails to demonstrate distinct academic research value or practical application significance. The study requires substantial improvement in justifying its theoretical contributions, technical innovation, and operational superiority over existing detection methods.
Author Response
Comments 1: [Representativeness of defect classification: The representativeness of the five categorized rail head surface defects requires scrutiny. These defects (all causing abrupt motion changes in the HRT robot) belong to surface geometric anomalies that typically generate significant vibrations during train operations. Current maintenance practices already detect such defects using conventional track inspection vehicles. The paper lacks justification for the necessity of employing HRT robots for precise detection and classification of these defects.]
Response 1:[Thank you very much for the professional suggestions given by the reviewers.
The 12 common surface damage defects of the rails analyzed and summarized in the article are classified into 5 major categories according to the characteristics of the detection methods and the damage. These 5 major categories of rail surface damage defects cover almost the vast majority of the surface damage defects of the rails, and there is a high probability of encountering them during the current inspection of high-speed railway lines, which shows certain representativeness.It is mainly restricted by the experimental test environment. The length of the rails used in the laboratory is limited. However, the verified inspection robot is completely designed and developed in accordance with the detection requirements of rail damage defects on the actual lines, and it is capable of detecting surface damage defects of the rails during high-speed operation. Therefore, the method proposed in the article has been experimentally verified within a certain speed range, and the accuracy rate of its detection of rail surface damage defects has been tested. In the next step, it will be tested on the actual lines. Moreover, relevant cooperation has been reached with the Chengdu Passenger Transport Section of Chengdu Railway Bureau, and field experimental tests will be carried out on the actual lines. The test speed will be based on the speed of the high-speed train tracks for experimental testing. This work will be the key research task in the next stage.
Thank you very much for the extremely professional comments and suggestions from the reviewers! Thank you!]
Comments 2:[Curves and gradients parameter exclusion: The study neglects to address how to distinguish and eliminate the influence of widespread track features like curves and gradients in actual track lines from the identified five defect categories.] Response 2:[Thank you very much for the professional suggestions provided by the reviewers. The damage defect of the slope is mainly manifested as a protrusion on the surface of the rail. In formula (1) of the article, a height h1 is reflected. When encountering a slope damage defect, the parameter height h1 will also change abruptly. However, if a horizontal curve damage defect is encountered, since there is no height change in the vertical direction for the horizontal curve damage defect, the value of the h1 parameter of the inspection robot at this time is 0. Based on this, it is possible to determine whether the damage defect at this location is a slope and gradient damage defect or a horizontal curve damage defect. Thank you very much for the extremely professional comments and suggestions from the reviewers! Thank you!] Comments 3:[High-speed railway relevance: No evident connection is established between the proposed defect classifications and their specific implications for high-speed railway operations.] Response 3:[Thank you very much for the professional suggestions given by the reviewers. The 12 types of surface damages of high-speed railway rails analyzed in this article are specifically focused on the surface damages of rails in high-speed railways. They are classified according to the damage types, and there are a total of 5 major categories. During the operation of high-speed railways, the regular maintenance and repair, as well as the detection and repair of surface damages of rails, are important tasks in the regular maintenance and repair work of high-speed railways. At the same time, the analysis and summary of the surface damage defects of the rails are mainly carried out in accordance with the requirements in the IRS 70712 and EN 17397-1:2020 standards, including the damage defects that need to be detected and so on. Thank you very much for the extremely professional comments and suggestions from the reviewers! Thank you!] Comments 4:[Robotic intelligence deficiency: The described HRT robot essentially functions as a conventional track inspection vehicle with mobility, without demonstrating intelligent features that would qualify it as a robotic system in terms of operational procedures or functionality.] Response 4:[Thank you very much for the professional suggestions provided by the reviewers. The designed and developed HRT robot is capable of automatically identifying and detecting the surface damages of the rails. It can also draw real-time status curves of the damage defects according to the detection results. As shown on the control computer page in Figure 1 of the article, it is the independently developed damage recognition system. Finally, the detected data of the damage defects will be stored in the database, and a large-scale damage model will be created to provide data support for subsequent operation and maintenance. Figure 2 of the article lists the hardware design of the HRT robot, which enables wireless remote control and automatically detects the damage defects of the rails. Thank you very much for the extremely professional comments and suggestions from the reviewers! Thank you!] Comments 5:[Academic depth limitation: The data analysis methods presented for the five defect types appear superficial, lacking sophisticated analytical approaches that would enhance academic rigor.] Response 5:[Thank you very much for the professional suggestions given by the reviewers. In this article, 12 common surface damage defects of rails are analyzed and summarized based on the IRS 70712 and EN 17397-1:2020 standards. Moreover, the HRT robot has been independently developed, and experimental tests have been carried out on the method proposed in the article. Probably mainly due to the limitations of the experimental test environment, although the experiments and verifications of the proposed method have been completed in combination with the independently developed HRT robot, and the results have met the expectations. However, in order to make the entire detection system more reliable and stable, in the next stage, the key breakthroughs will be focused on aspects such as the detection algorithm and method for rail damage defects, including the simultaneous detection of multiple damage defects. This also includes relevant experimental tests and verifications, as well as conducting experimental tests and verifications in combination with actual railway lines to further highlight the depth of the research. This work will be the key research task in the next stage. Thank you very much for the extremely professional comments and suggestions from the reviewers! Thank you!]
Reviewer 3 Report (New Reviewer)
Comments and Suggestions for Authors
The author introduces a fault diagnosis method in the field of steel rails, which has certain value. The author needs further revisions to make it a mature publication.
1. The chart should be further clarified to ensure that the formula symbols are correct and standardized.
2. The introduction further emphasizes the advantages of design motivation and solutions.
3. Some effective deep learning fault diagnosis ideas should be emphasized in the introduction, such as An X-Ray-Based Multiexpert Inspection Method for Automatic Welding Defect, A Novel Fault Diagnosis Method of High-Speed Train Based on Few-Shot Learning,A Reliable Bolt Key-Points Detection Method in Industrial Magnetic Separator Systems。 Damage detection for high-speed railway standard box girders based on time–frequency characteristics of train-induced strain
4. Further discussion and analysis of limitations are necessary. The author has the responsibility to explain under what circumstances the use of this method is effective.
5. Case studies of real-world application scenarios can be further discussed to increase practicality.
Overall, this article is solid and requires some modifications that can be accepted.
Author Response
Comments 1: [The chart should be further clarified to ensure that the formula symbols are correct and standardized.]
Response 1:[Thank you very much for the professional suggestions provided by the reviewers.
The formulas and other elements in the charts have already been revised and supplemented in the article.
Thank you very much for the extremely professional comments and suggestions from the reviewers! Thank you!]
Comments 2: [The introduction further emphasizes the advantages of design motivation and solutions.]
Response 2:[Thank you very much for the professional suggestions given by the reviewers.
Revisions and supplements have been made to the introduction part of the literature. The two paragraphs of text are indicated in red font. Thank you very much for the extremely professional comments and suggestions from the reviewers! Thank you!]
Comments 3:[Some effective deep learning fault diagnosis ideas should be emphasized in the introduction, such as An X-Ray-Based Multiexpert Inspection Method for Automatic Welding Defect, A Novel Fault Diagnosis Method of High-Speed Train Based on Few-Shot Learning,A Reliable Bolt Key-Points Detection Method in Industrial Magnetic Separator Systems。 Damage detection for high-speed railway standard box girders based on time–frequency characteristics of train-induced strain]
Response 3:[Thank you very much for the professional suggestions provided by the reviewers.
Revisions and supplements have been made to the introduction part of the literature. The two paragraphs of text are indicated in red font. Three pieces of literature have also been added to the reference list. Thank you very much for the extremely professional comments and suggestions from the reviewers! Thank you!]
Comments 4:[Further discussion and analysis of limitations are necessary. The author has the responsibility to explain under what circumstances the use of this method is effective.]
Response 4:[Thank you very much for the professional suggestions provided by the reviewers.
The method proposed in this article is mainly analyzed according to the requirements in the IRS 70712 and EN 17397-1:2020 standards, including the damage defects that need to be detected. Therefore, it is effective for the detection of the existing surface damage defects of high-speed railway rails.This has been supplemented in the conclusion part at the end of the article. Thank you very much for the extremely professional comments and suggestions from the reviewers! Thank you!]
Comments 5:[Case studies of real-world application scenarios can be further discussed to increase practicality.]
Response 5:[Thank you very much for the professional suggestions provided by the reviewers.
It has been supplemented in the conclusion part at the end of the article. Thank you very much for the extremely professional comments and suggestions from the reviewers! Thank you!]
Round 2
Reviewer 2 Report (New Reviewer)
Comments and Suggestions for Authors
The revised manuscript fails to address the reviewers' key concerns regarding research justification and academic contributions. Notably, it lacks substantial improvements in the following critical aspects:
-
Unresolved research necessity: The study still inadequately justifies the academic significance and practical urgency of developing HRT robots, given that their functional performance shows no breakthrough compared to existing track inspection vehicles.
-
Academic value deficiency: No substantial theoretical innovation or technical advancement has been demonstrated in either the robotic system design or defect detection methodology.
-
Limited application potential: The proposed HRT robot exhibits no measurable performance superiority over conventional inspection vehicles in terms of detection accuracy, operational efficiency, or adaptability to complex track environments (e.g., curves and slopes).
These unresolved issues fundamentally undermine the manuscript's eligibility for publication, as it currently offers neither novel scientific insights nor demonstrable engineering value beyond established industry practices.
Author Response
Please see the attachment.

This manuscript is a resubmission of an earlier submission. The following is a list of the peer review reports and author responses from that submission.
Round 1
Reviewer 1 Report
Comments and Suggestions for Authors
This paper proposes a rail head damage detection method based on the motion posture of an inspection robot. While the concept is interesting, the study lacks the necessary depth and rigor to meet publication standards. Therefore, I regret to inform you that I must reject this paper. My detailed review comments are as follows:
- The proposed method appears to be specifically designed for detecting defects on the rail head surface. This severely limits its potential applicability. Have the authors considered whether this approach is suitable for detecting defects on the rail web or rail foot?
- The title emphasizes both the robotic system and the fault detection method. However, after reviewing the content, I find that the study merely presents a low Technology Readiness Level (TRL) feasibility verification rather than a fully developed robotic system. The term "robot" in the title is therefore misleading.
- Line 35: There is a spelling error. "seven people's"?
- Line 79: The wording is ambiguous. The phrase "used eddy current testing technology to manually push the detection vehicle" lacks clarity. Does it imply that eddy current testing was applied while manually pushing the vehicle?
- Line 119: The word "research" appears redundant. Please verify its necessity in the sentence.
- In Section 2.1, if the authors consider this system to be a "robot," they should specify its operational endurance (e.g., battery life and range).
- Line 163: The authors claim that the system can achieve a maximum operating speed of 20 m/s (equivalent to 72 km/h). I find this highly questionable, as the current design does not appear capable of reaching such a high speed while performing inspections.
- In Figure 1, multiple sensors are visibly placed on the rail web. However, the authors provide no details regarding the type or function of these sensors. What are their specifications, and what role do they play in the system?
- Figure 2 is extremely blurry, making it difficult to discern its details. Additionally, the paper mentions the use of both 3D point clouds and 2D images for defect detection. However, there is no explanation of the data collected by these sensors or how they contribute to the detection process. If the system integrates these modalities, why is there no discussion on data fusion between the motion posture module and these sensor data?
- Line 183: The paper states that Wi-Fi is used to control the robot. While this may be feasible in a laboratory setting, it is highly impractical in a real railway environment due to connectivity limitations.
- In Section 3.1, the classification of rail defects needs further refinement. The authors seem to consider only 2D defect scenarios, where all defects appear to through the entire rail head. However, many real-world defects—such as Rolling Contact Fatigue (RCF) defects and Rail Head Squats—are localized and do not necessarily extend across the entire rail head. Moreover, some of these defects may not significantly impact the motion posture of the system, yet they are among the most common in practice. The authors fail to account for these important defect types.
- In Section 3.2, the authors demonstrate a misunderstanding of kinematic analysis of robots. The title suggests that the section should focus on the kinematics of different robot components, such as the front and rear wheels and sensors. However, the authors treat the entire system as a single rigid body and analyze only its overall acceleration, velocity, and displacement. This is a significant research flaw. During obstacle traversal, different parts of the system will experience different accelerations and velocities. Furthermore, in the provided equations, the authors seem to assume a constant acceleration, which is unrealistic in practical scenarios.
- Lines 291–292: Please provide appropriate references to support the claims made in these lines.
- In Section 3.2, regarding the motion posture of the system, I strongly recommend measuring angular velocity and angular acceleration rather than relying solely on linear velocity and linear acceleration. The latter approach is highly inaccurate. Additionally, the authors fail to explain why the system consistently exhibits acceleration while its velocity remains largely unchanged. Is this due to sensor noise or offset errors?
- In Section 4, Line 404, the reported average testing speed is only 08 m/s (equivalent to 0.288 km/h). This contradicts the authors' repeated claims about "high-efficiency inspection." Moreover, the authors do not specify the size of the detected defects. At such a low speed, the motion posture data obtained has little practical relevance. When speed increases, the variations in acceleration and velocity caused by small defects may be entirely masked by environmental disturbances and noise. I strongly recommend conducting tests at different speeds and analysing the collected data accordingly, especially considering the authors' claim that the system can operate at 72 km/h.
- In Figures 19 and 20, the x-axis labelling is unclear. What does "s/m" represent? This notation is ambiguous and needs clarification.
Overall, while the paper presents an intriguing idea, it lacks a thorough and rigorous analysis. Many claims remain unsubstantiated, and critical methodological details are missing. Significant improvements are necessary before this work can be considered for publication.
Author Response
Comments 1: [The proposed method appears to be specifically designed for detecting defects on the rail head surface. This severely limits its potential applicability. Have the authors considered whether this approach is suitable for detecting defects on the rail web or rail foot?]
Response 1: [Thank you to the review experts for their valuable comments. The method proposed in this article can identify and detect surface damages on the rail head, web, and base. Therefore, it can be applied to detect surface damages on the rail web and base. This article focuses on the analysis and experimental verification of surface damages on the rails.]Thank you for pointing this out. I/We agree with this comment.
Comments 2: [The title emphasizes both the robotic system and the fault detection method. However, after reviewing the content, I find that the study merely presents a low Technology Readiness Level (TRL) feasibility verification rather than a fully developed robotic system. The term "robot" in the title is therefore misleading.]
Response 2: [Thank you to the review experts for their valuable comments. The inspection robot used for testing in this article is completely independently designed and developed, covering both the hardware and software aspects. The assembly of the hardware part is shown in Figure 1 of the article, and Figure 2 presents the physical logic diagram of the robot's sensors and control system. The HRT robot is independently designed and developed, and six patents and two software copyrights have been applied for, including the upper computer software and APP that can be wirelessly controlled via WiFi, all of which are independently designed and developed. Since the main focus of this article is on the experimental testing of rail damage inspection, this part of the content has not been elaborated in detail in this article. Regarding the design and development of the inspection robot, the relevant technical content is currently being organized and will be submitted as another academic paper.]
Comments 3: [Line 35: There is a spelling error. "seven people's"?]
Response 3:[Thank you for the valuable comments from the review experts. The article has been revised. The revised content is as follows: The use of rail transit can be traced back to ancient Greece in the 6th century BC at the earliest.]Thank you for pointing this out. I/We agree with this comment.
Comments 4: [Line 79: The wording is ambiguous. The phrase "used eddy current testing technology to manually push the detection vehicle" lacks clarity. Does it imply that eddy current testing was applied while manually pushing the vehicle?]
Response 4:[Thank you for the valuable comments from the review experts. The relevant part in the article has been revised. The revised content is: The manually pushed rail inspection vehicle developed by Sperry Corporation in the United States is based on eddy current testing technology.]Thank you for pointing this out. I/We agree with this comment.
Comments 5: [Line 119: The word "research" appears redundant. Please verify its necessity in the sentence.]
Response 5:[Thank you for the valuable comments from the review experts. The article has been revised. It has been deleted.]Thank you for pointing this out. I/We agree with this comment.
Comments 6: [In Section 2.1, if the authors consider this system to be a "robot," they should specify its operational endurance (e.g., battery life and range).]
Response 6:[Thank you for the valuable comments from the review experts. The relevant part in the article has been revised. The revised content is: When the power supply is normal, the robot's endurance time is 7 to 8 hours.]Thank you for pointing this out. I/We agree with this comment.
Comments 7:[Line 163: The authors claim that the system can achieve a maximum operating speed of 20 m/s (equivalent to 72 km/h). I find this highly questionable, as the current design does not appear capable of reaching such a high speed while performing inspections.]
Response 7:[Thank you for the valuable comments from the review experts. This speed is the theoretical design calculation speed value under ideal conditions, and the actual running speed should be lower than this value, which requires testing on the actual line. The relevant part in the article has been revised. The revised content is: Under ideal conditions, the maximum speed calculated based on the theoretical design can reach 20 m/s.]
Comments 8:[In Figure 1, multiple sensors are visibly placed on the rail web. However, the authors provide no details regarding the type or function of these sensors. What are their specifications, and what role do they play in the system?]
Response 8:[Thank you for the valuable comments from the review experts. The relevant content such as the parameters of the sensor has been supplemented, and the article has been revised accordingly. The revised content is: The sensor is an X-Y-Z three-axis piezoelectric acceleration sensor of the model 1A339E. It has a sensitivity of 5mV/m² and can collect signals within a frequency range of 2-10000Hz.]Thank you for pointing this out. I/We agree with this comment.
Comments 9:[Figure 2 is extremely blurry, making it difficult to discern its details. Additionally, the paper mentions the use of both 3D point clouds and 2D images for defect detection. However, there is no explanation of the data collected by these sensors or how they contribute to the detection process. If the system integrates these modalities, why is there no discussion on data fusion between the motion posture module and these sensor data?]
Response 9:[Thank you for the valuable comments from the review experts. This part of the content was redundant and has been deleted. The article has been revised accordingly. It has been deleted.Figure 2 has been revised.]Thank you for pointing this out. I/We agree with this comment.
Comments 10:[Line 183: The paper states that Wi-Fi is used to control the robot. While this may be feasible in a laboratory setting, it is highly impractical in a real railway environment due to connectivity limitations.]
Response 10:[Thank you for the valuable comments from the review experts. The WiFi - based control method still requires further experimental verification on the actual line. When conducting tests on the actual line, GPS will be given priority as it is more stable than WiFi and is also the control and positioning method for the next - generation intelligent inspection robot control system. In the article, the experimental tests were carried out using WiFi control, which is feasible in the experimental environment.]Thank you for pointing this out. I/We agree with this comment.
Comments 11:[In Section 3.1, the classification of rail defects needs further refinement. The authors seem to consider only 2D defect scenarios, where all defects appear to through the entire rail head. However, many real-world defects—such as Rolling Contact Fatigue (RCF) defects and Rail Head Squats—are localized and do not necessarily extend across the entire rail head. Moreover, some of these defects may not significantly impact the motion posture of the system, yet they are among the most common in practice. The authors fail to account for these important defect types.]
Response 11:[Thank you for the valuable comments from the review experts. The relevant tests mentioned in the article are about the surface damage of the rails, which includes the local damage cracks on the rail surface. The crushed pits, inclined cracks, etc. mentioned in the article are all analyzed for the local damage on the rail surface.]
Comments 12:[In Section 3.2, the authors demonstrate a misunderstanding of kinematic analysis of robots. The title suggests that the section should focus on the kinematics of different robot components, such as the front and rear wheels and sensors. However, the authors treat the entire system as a single rigid body and analyze only its overall acceleration, velocity, and displacement. This is a significant research flaw. During obstacle traversal, different parts of the system will experience different accelerations and velocities. Furthermore, in the provided equations, the authors seem to assume a constant acceleration, which is unrealistic in practical scenarios.]
Response 12:[Thank you for the valuable comments from the review experts. When detecting the movement of the inspection robot on the rail surface, the robot system is regarded as a whole, a rigid - body system. Also, based on the physical properties of the inspection robot system, data collection and analysis are carried out for the entire rigid - body system. Precisely because it is regarded as a rigid - body system, when the robot crosses an obstacle, the overall rigid - body system has a certain instantaneous velocity, acceleration, and displacement. And under different motion states, it shows different values, making the final test results closer to reality. When analyzing the wheel - rail relationship of high - speed train carriages, the carriage is also regarded as a rigid - body system. When conducting a coupled analysis of the vehicle and the track, it is also analyzed as a whole rigid - body system. During the testing process, the acceleration value in the formula is different for different motion states. At the same time, as can be seen from the test result figures 8 - 17, the acceleration values are different under different motion states.]Thank you for pointing this out. I/We agree with this comment.
Comments 13:[Lines 291–292: Please provide appropriate references to support the claims made in these lines.]
Response 13:[Thank you for the valuable comments from the review experts. Relevant literature has been supplemented and added.]Thank you for pointing this out. I/We agree with this comment.
Comments 14:[In Section 3.2, regarding the motion posture of the system, I strongly recommend measuring angular velocity and angular acceleration rather than relying solely on linear velocity and linear acceleration. The latter approach is highly inaccurate. Additionally, the authors fail to explain why the system consistently exhibits acceleration while its velocity remains largely unchanged. Is this due to sensor noise or offset errors?]
Response 14:[Thank you for the valuable comments from the review experts. During the testing process of the inspection robot, when passing through damage defects, the robot's instantaneous acceleration and speed both change suddenly. As can be seen from the test result curve, the moment of the sudden change can be identified. During the wheel - rail contact process, vibration is generated due to the wheel - rail contact, so vibration acceleration is produced. In actual lines, when high - speed trains are running, vibration is generated when the wheels contact the rails, and vibration acceleration always exists. This is mainly caused by the vibration generated during wheel - rail contact.]
Comments 15:[In Section 4, Line 404, the reported average testing speed is only 08 m/s (equivalent to 0.288 km/h). This contradicts the authors' repeated claims about "high-efficiency inspection." Moreover, the authors do not specify the size of the detected defects. At such a low speed, the motion posture data obtained has little practical relevance. When speed increases, the variations in acceleration and velocity caused by small defects may be entirely masked by environmental disturbances and noise. I strongly recommend conducting tests at different speeds and analysing the collected data accordingly, especially considering the authors' claim that the system can operate at 72 km/h.]
Response 15:[Thank you for the valuable comments from the review experts. In the article, the defect damages are classified, and tests are carried out on five types of damage defects. The numerical deterioration parameters of the five types of damage defects are all explained. For example, "As shown in Figure 14, the motion attitude parameters of the HRT robot when the gap distances are 0.5mm, 1.5mm, and 2.5mm respectively." The parameter sizes of the damage defects are all stated in the classification tests. The speed in the tests is the speed during laboratory tests. Due to the limitations of experimental conditions, further tests and verifications will be carried out on the actual line later, and tests will be conducted at different speeds.]
Comments 16:[In Figures 19 and 20, the x-axis labelling is unclear. What does "s/m" represent? This notation is ambiguous and needs clarification.]
Response 16:[Thank you for the valuable comments from the review experts. Supplementary explanations have been added to the article. The added content is: In Figures 19 and 20, the horizontal axis represents the displacement distance of the robot. The physical quantity of the displacement distance is denoted by "s", and the unit is meter (m).]Thank you for pointing this out. I/We agree with this comment.
Reviewer 2 Report
Comments and Suggestions for Authors
This paper presents a case study of developing a new rail damage inspection robot and defect detection method for high-speed railways. However, there is insufficient information on the performance criteria and limitations of the robot’s defect detection. Therefore, in order to secure the accuracy of defect detection for rail damage inspection, we ask that you review the following additional modifications.
- Please explain specifically why you inserted the word ‘high-speed’ in the title of this paper.
- Since only the types of rail damage are presented in this paper, it is necessary to clearly present performance criteria and limitations such as the size of the defect in order to prove the superiority of this robot.
- The p.15 HRT robot was tested twice, and after passing through five types of defects, please record the changes in the acceleration and speed of the HRT robot’s motion and present the results of a statistical analysis. It is thought that it would be difficult to obtain statistical analysis results with only two tests. In addition, the measurement uncertainty of this equipment according to each type of defect should be statistically organized.
- The x-axis description in Figure 19 is missing. Please explain the cause of the difference between TC-Measurement1 and 2.
- On lines 248-249 of page 8, the words Desending Step Obstacle Movoement are capitalized even though they are not abbreviations. On other pages, the words presented with the Defect name are presented in lowercase.
Please write the expressions for words by distinguishing between uppercase and lowercase letters. - Please summarize the quantitative research results in the conclusion section and increase the size and resolution of all figures.
Author Response
Comments 1: [Please explain specifically why you inserted the word ‘high-speed’ in the title of this paper.]
Response 1: [Thank you for the valuable comments from the review experts. High - speed railway trains run at a very fast speed. During operation, the failure rate of contact damage between train wheels and the rail surface is relatively high. At present, the maintenance of high - speed railway tracks mainly relies on manual inspection. Moreover, compared with ordinary - speed railways, the damage detection of high - speed railway tracks not only has a higher failure rate but also greater inspection difficulty. Therefore, the original intention of the design mainly focuses on the damage detection of high - speed railway tracks, so the term "high - speed railway" is written in the title. The tests carried out in the laboratory are mainly limited by the indoor conditions of the laboratory. In the later stage, tests will be carried out on the line, and the test speed will be tested according to the designed speed.]
Comments 2: [Since only the types of rail damage are presented in this paper, it is necessary to clearly present performance criteria and limitations such as the size of the defect in order to prove the superiority of this robot.]
Response 2: [Thank you for the valuable comments from the review experts. In the article, the defect damages are classified, and tests are conducted on five types of damage defects. The numerical deterioration parameters of these five types of damage defects are all explained. For example, "As shown in Figure 14, the motion attitude parameters of the HRT robot when the gap distances are 0.5mm, 1.5mm, and 2.5mm respectively." The parameter values of the damage defects are all specified in the classification tests.]
Comments 3:[The p.15 HRT robot was tested twice, and after passing through five types of defects, please record the changes in the acceleration and speed of the HRT robot’s motion and present the results of a statistical analysis. It is thought that it would be difficult to obtain statistical analysis results with only two tests. In addition, the measurement uncertainty of this equipment according to each type of defect should be statistically organized.]
Response 3: [Thank you for the valuable comments from the review experts. The two tests conducted at the end are intended to serve as a positive comparison for the previous tests of the five types of damage defects, providing positive data support and creating a contrast. The actual damage - related test detections are part of the previous tests of the five types of damage defects.We sincerely appreciate the guidance of the review experts. When conducting tests on the actual line in the future, a more comprehensive test plan will be formulated to conduct multi - perspective tests on different types of rail damage.]
Comments 4:[The x-axis description in Figure 19 is missing. Please explain the cause of the difference between TC-Measurement1 and 2.]
Response 4: [Thank you for the valuable comments from the review experts. The description of the meaning of the X - axis in Figure 19 has been added and incorporated into the article.TC - type damage refers to groove - type damage. Each time the robot crosses a TC - type damage defect, the vibration between the wheel - rail contact and the height difference of the groove damage defect itself will cause the acceleration and speed to change suddenly in an instant. Since there is inertia during the robot's operation, when crossing the TC - groove - type damage defect, the acceleration and speed experience a jump - like sudden change.]
Comments 5:[On lines 248-249 of page 8, the words Desending Step Obstacle Movoement are capitalized even though they are not abbreviations. On other pages, the words presented with the Defect name are presented in lowercase.
Please write the expressions for words by distinguishing between uppercase and lowercase letters.]
Response 5: [Thank you for the valuable comments from the review experts. It has been revised and the relevant content has been added to the article.]
Comments 6:[Please summarize the quantitative research results in the conclusion section and increase the size and resolution of all figures.]
Response 6: [Thank you for the valuable comments from the review experts. It has been revised and the relevant content has been added to the article.]
Reviewer 3 Report
Comments and Suggestions for Authors
Your paper is very nice and interesting. The investigations on the railway surface is very important for the maintenance service. What I did not notice is that are no references regarding the damages of the rail structure or the curve elevation. Also will be interesting to write something about the robot capacity (how much can inspect/night- I assume that this inspections are made exclusively by night; there are some limitations regarding the weather?) and if it is necessary the human assistance. The experimental speed in my opinion is too low (0.8m/s vs 20m/s that you declare that it can do). The robot must be also tested on a real scale railway- because the 2 figures Fig.19-20 indicates a lot of damage of the inspected rail way- at a very low speed 0.08m/sec=aprox.0.3km/h. If you can not present now a real rail way sector you must make at the end of the paper the future results and investigations that you will make to validate the robot.
Author Response
Comments 1: [Your paper is very nice and interesting. The investigations on the railway surface is very important for the maintenance service. What I did not notice is that are no references regarding the damages of the rail structure or the curve elevation. Also will be interesting to write something about the robot capacity (how much can inspect/night- I assume that this inspections are made exclusively by night; there are some limitations regarding the weather?) and if it is necessary the human assistance. The experimental speed in my opinion is too low (0.8m/s vs 20m/s that you declare that it can do). The robot must be also tested on a real scale railway- because the 2 figures Fig.19-20 indicates a lot of damage of the inspected rail way- at a very low speed 0.08m/sec=aprox.0.3km/h. If you can not present now a real rail way sector you must make at the end of the paper the future results and investigations that you will make to validate the robot.]
Response 1: [Thank you for the valuable comments from the review experts. Papers on rail result damage have been supplemented and added, as shown in References 3 - 4. Regarding the actual detection capabilities of the robot, the tests of the inspection robot in the laboratory for the detection of rail surface damage have been completed. The next step will be to conduct tests on the actual line, which is the work to be done next. Due to the limitations of experimental conditions in the laboratory, the classification tests of damage have been completed at present, and all have achieved ideal results. Further relevant experimental tests will be carried out later, and tests will be conducted on the line. We are extremely grateful for the valuable comments of the review experts. Meanwhile, the review experts have further guided the relevant research work and provided constructive suggestions. Thank you very much to the review experts.]
Round 2
Reviewer 1 Report
Comments and Suggestions for Authors
Dear Authors,
Thank you for your revised manuscript and response letter. After carefully reviewing the revisions and your replies, I find that the manuscript and the research work still require substantial improvements. Therefore, I regret to inform you that I must recommend its rejection.
Comments 1: [The proposed method appears to be specifically designed for detecting defects on the rail head surface. This severely limits its potential applicability. Have the authors considered whether this approach is suitable for detecting defects on the rail web or rail foot?]
Response 1: [Thank you to the review experts for their valuable comments. The method proposed in this article can identify and detect surface damages on the rail head, web, and base. Therefore, it can be applied to detect surface damages on the rail web and base. This article focuses on the analysis and experimental verification of surface damages on the rails.]Thank you for pointing this out. I/We agree with this comment.
Comment: I still find it unclear how defects on the rail web or foot surface influence the motion posture of the robot, given that the robot’s wheels only make contact with the top surface of the rail head. The explanation provided does not adequately address this concern, and I remain unsatisfied with this response.
Comments 2: [The title emphasizes both the robotic system and the fault detection method. However, after reviewing the content, I find that the study merely presents a low Technology Readiness Level (TRL) feasibility verification rather than a fully developed robotic system. The term "robot" in the title is therefore misleading.]
Response 2: [Thank you to the review experts for their valuable comments. The inspection robot used for testing in this article is completely independently designed and developed, covering both the hardware and software aspects. The assembly of the hardware part is shown in Figure 1 of the article, and Figure 2 presents the physical logic diagram of the robot's sensors and control system. The HRT robot is independently designed and developed, and six patents and two software copyrights have been applied for, including the upper computer software and APP that can be wirelessly controlled via WiFi, all of which are independently designed and developed. Since the main focus of this article is on the experimental testing of rail damage inspection, this part of the content has not been elaborated in detail in this article. Regarding the design and development of the inspection robot, the relevant technical content is currently being organized and will be submitted as another academic paper.]
Comment: Since the primary focus of this article is on the experimental testing of rail damage inspection, the title should place greater emphasis on the inspection method rather than on 'robot,' which appears to serve more as a technology verification platform.
Comments 8:[In Figure 1, multiple sensors are visibly placed on the rail web. However, the authors provide no details regarding the type or function of these sensors. What are their specifications, and what role do they play in the system?]
Response 8:[Thank you for the valuable comments from the review experts. The relevant content such as the parameters of the sensor has been supplemented, and the article has been revised accordingly. The revised content is: The sensor is an X-Y-Z three-axis piezoelectric acceleration sensor of the model 1A339E. It has a sensitivity of 5mV/m² and can collect signals within a frequency range of 2-10000Hz.]Thank you for pointing this out. I/We agree with this comment.
Comment: The authors' response is even more confusing. I initially understood that the accelerometers were mounted on the robot to monitor its motion posture. However, the authors have instead mounted them on the rail web without providing any explanation. Given that the accelerometer is presumably a part of the robot's sensor module, it would be logical for it to be installed on the robot itself. Could the authors clarify this discrepancy?
Comments 9:[Figure 2 is extremely blurry, making it difficult to discern its details. Additionally, the paper mentions the use of both 3D point clouds and 2D images for defect detection. However, there is no explanation of the data collected by these sensors or how they contribute to the detection process. If the system integrates these modalities, why is there no discussion on data fusion between the motion posture module and these sensor data?]
Response 9:[Thank you for the valuable comments from the review experts. This part of the content was redundant and has been deleted. The article has been revised accordingly. It has been deleted.Figure 2 has been revised.]Thank you for pointing this out. I/We agree with this comment.
Comment: The authors have still not provided a clear explanation regarding the use of both 3D point clouds and 2D images for defect detection. There is no detailed discussion on the data collected by these sensors or how they contribute to the detection process.
Additionally, I noticed that Figure 2 includes a sensor for monitoring acceleration. I believe this sensor is the appropriate module for defect detection. Therefore, the authors should clarify the functional differences between this sensor and the sensors mounted on the rail web.
Comments 11:[In Section 3.1, the classification of rail defects needs further refinement. The authors seem to consider only 2D defect scenarios, where all defects appear to through the entire rail head. However, many real-world defects—such as Rolling Contact Fatigue (RCF) defects and Rail Head Squats—are localized and do not necessarily extend across the entire rail head. Moreover, some of these defects may not significantly impact the motion posture of the system, yet they are among the most common in practice. The authors fail to account for these important defect types.]
Response 11:[Thank you for the valuable comments from the review experts. The relevant tests mentioned in the article are about the surface damage of the rails, which includes the local damage cracks on the rail surface. The crushed pits, inclined cracks, etc. mentioned in the article are all analyzed for the local damage on the rail surface.]
Comment: In Table 12, are all the defect images presented as side views? Additionally, are PCD and VCD top-view schematic diagrams? Furthermore, many small defects in reality, such as minor rail squats, may not impact the robot's motion posture. However, the authors have only listed defects that could potentially influence the robot's motion while neglecting others. A more comprehensive analysis, including defects that do not affect motion posture, would provide a more complete evaluation of the inspection method.
Comments 13:[Lines 291–292: Please provide appropriate references to support the claims made in these lines.]
Response 13:[Thank you for the valuable comments from the review experts. Relevant literature has been supplemented and added.]Thank you for pointing this out. I/We agree with this comment.
Comment: No reference added.
Comments 14:[In Section 3.2, regarding the motion posture of the system, I strongly recommend measuring angular velocity and angular acceleration rather than relying solely on linear velocity and linear acceleration. The latter approach is highly inaccurate. Additionally, the authors fail to explain why the system consistently exhibits acceleration while its velocity remains largely unchanged. Is this due to sensor noise or offset errors?]
Response 14:[Thank you for the valuable comments from the review experts. During the testing process of the inspection robot, when passing through damage defects, the robot's instantaneous acceleration and speed both change suddenly. As can be seen from the test result curve, the moment of the sudden change can be identified. During the wheel - rail contact process, vibration is generated due to the wheel - rail contact, so vibration acceleration is produced. In actual lines, when high - speed trains are running, vibration is generated when the wheels contact the rails, and vibration acceleration always exists. This is mainly caused by the vibration generated during wheel - rail contact.]
Comment: I literally understand the statement that 'during the wheel-rail contact process, vibration is generated due to the wheel-rail contact.' However, the vibration should fluctuate both positively and negatively, with a mean value tending toward zero, rather than exhibiting the offset curves shown in the figures. In the figures, the acceleration curves appear to be predominantly positive, which suggests an increase in speed. However, the speed curves remain relatively constant with fluctuations, rather than showing a corresponding increase. Could the authors clarify this discrepancy?
Additionally, the authors mentioned that the sensor used is a three-axis piezoelectric acceleration sensor. However, the figure does not indicate the direction of measurement. Can the authors specify the orientation of the sensor as shown in the figure? Furthermore, is it possible to extract vibration features from all three directions?
Comments 15:[In Section 4, Line 404, the reported average testing speed is only 08 m/s (equivalent to 0.288 km/h). This contradicts the authors' repeated claims about "high-efficiency inspection." Moreover, the authors do not specify the size of the detected defects. At such a low speed, the motion posture data obtained has little practical relevance. When speed increases, the variations in acceleration and velocity caused by small defects may be entirely masked by environmental disturbances and noise. I strongly recommend conducting tests at different speeds and analysing the collected data accordingly, especially considering the authors' claim that the system can operate at 72 km/h.]
Response 15:[Thank you for the valuable comments from the review experts. In the article, the defect damages are classified, and tests are carried out on five types of damage defects. The numerical deterioration parameters of the five types of damage defects are all explained. For example, "As shown in Figure 14, the motion attitude parameters of the HRT robot when the gap distances are 0.5mm, 1.5mm, and 2.5mm respectively." The parameter sizes of the damage defects are all stated in the classification tests. The speed in the tests is the speed during laboratory tests. Due to the limitations of experimental conditions, further tests and verifications will be carried out on the actual line later, and tests will be conducted at different speeds.]
Comment: The low operational speed used in this study is literally neither practical nor acceptable. The authors must carefully select experimental parameters that accurately represent real-world conditions when validating their proposed technologies. From this perspective, the Design of Experiments (DoE) is severely inadequate.
Moreover, the authors have not accounted for environmental noise and disturbances that may arise at higher operational speeds. The validation conducted at the current test speed lacks rigor and fails to adequately demonstrate the feasibility of defect detection based on the robot’s motion posture.
Author Response
Comments 1:[I still find it unclear how defects on the rail web or foot surface influence the motion posture of the robot, given that the robot’s wheels only make contact with the top surface of the rail head. The explanation provided does not adequately address this concern, and I remain unsatisfied with this response]
Response 1: [Thank you for the valuable comments from the review experts. I'm extremely sorry for the unclear expression on my part, which caused your misunderstanding. Regarding the damages to the rail web and rail base, they cannot be detected by analyzing the motion state of the inspection robot and other data methods need to be combined. The new inspection method proposed in the article is mainly applied to the identification and detection of surface damages on the rail head. The full text has been revised to make the expression more rigorous. Thank you very much for the comments from the expert teacher!]
Comments 2:[Since the primary focus of this article is on the experimental testing of rail damage inspection, the title should place greater emphasis on the inspection method rather than on 'robot,' which appears to serve more as a technology verification platform.]
Response 2:[Thank you very much for the suggestions from the expert teacher. The title of the article has been revised to:A Novel Rail damage Fault Detection Method for high-speed railway.]
Comments 3:[The authors' response is even more confusing. I initially understood that the accelerometers were mounted on the robot to monitor its motion posture. However, the authors have instead mounted them on the rail web without providing any explanation. Given that the accelerometer is presumably a part of the robot's sensor module, it would be logical for it to be installed on the robot itself. Could the authors clarify this discrepancy?]
Response 3:[Thank you very much for the comments from the expert teacher. I didn't express myself clearly before. All the sensors are installed on the built test robot platform. It is precisely because they are installed on this platform that when the robot is in operation, the surface damage of the rail will cause changes in the motion state of the robot. The sensors can accurately collect the motion state data for damage identification. I'm really sorry for not making it clear and causing your misunderstanding. Thank you very much, expert teacher, for your professional and valuable comments!]
Comments 4:[The authors have still not provided a clear explanation regarding the use of both 3D point clouds and 2D images for defect detection. There is no detailed discussion on the data collected by these sensors or how they contribute to the detection process.]
Response 4:[Thank you very much for the comments from the expert teacher. The detection method proposed in the article mainly identifies the surface damage of the rail head by analyzing the motion state of the inspection robot. The 2D image sensors and 3D point cloud sensors mentioned in the article mainly reflect the structure and components of the built damage inspection robot. The full text has been revised, and the content related to the analysis of 2D images, 3D point clouds, etc. has been modified. Thank you very much, expert teacher, for your professional and valuable comments!]
Comments 5:[Additionally, I noticed that Figure 2 includes a sensor for monitoring acceleration. I believe this sensor is the appropriate module for defect detection. Therefore, the authors should clarify the functional differences between this sensor and the sensors mounted on the rail web.]
Response 5:[Thank you very much for the comments from the expert teacher. The sensor at the rail web in Figure 2 is solely used to collect the vibration acceleration data of the rail, and the collected data serves as backup data. The collected data is used for other purposes and has nothing to do with the content of this article. When taking photos after the experiment was completed, I forgot to unplug it.]
Comments 6:[In Table 12, are all the defect images presented as side views? Additionally, are PCD and VCD top-view schematic diagrams? Furthermore, many small defects in reality, such as minor rail squats, may not impact the robot's motion posture. However, the authors have only listed defects that could potentially influence the robot's motion while neglecting others. A more comprehensive analysis, including defects that do not affect motion posture, would provide a more complete evaluation of the inspection method. ]
Response 6:[Thank you very much for the comments from the expert teacher. In the table, ID, LBD, RBD, PCD, VCD, and ACD are top views, and the rest are side views. PCD and VCD are top views. What the expert teacher said is extremely correct and comprehensive. There are many very small damages on actual lines, such as fine crack damages. The detection of such minor damages still requires the combination of other methods to potentially identify them. The method presented in the article can identify some damages within a certain range, including a total of 12 common types of damage defects on the rail head for classification.
For very small damage defects, it can be improved in terms of enhancing the detection and recognition accuracy, or identified by combining other sensors and methods. This is also the direction and content of the research to be carried out in the next stage. Thank you very much for the expert teacher's guiding suggestions.]
Comments 7:[No reference added.]
Response 7:[Thank you very much for the comments from the expert teacher. The relevant literature content has been revised and added.]
Comments 8:[I literally understand the statement that 'during the wheel-rail contact process, vibration is generated due to the wheel-rail contact.' However, the vibration should fluctuate both positively and negatively, with a mean value tending toward zero, rather than exhibiting the offset curves shown in the figures. In the figures, the acceleration curves appear to be predominantly positive, which suggests an increase in speed. However, the speed curves remain relatively constant with fluctuations, rather than showing a corresponding increase. Could the authors clarify this discrepancy?]
Response 8:[Thank you very much for the comments from the expert teacher. The test robot platform conducts uniform - speed tests at different speeds. When it encounters surface damage defects on the rail head, the motion state of the test robot changes, the acceleration changes suddenly, and at the same time, the speed also changes suddenly. Correspondingly, in the figure, the acceleration suddenly increases and the corresponding speed also suddenly increases. As the expert teacher said, when the motion state changes suddenly, from the figure, when the acceleration changes suddenly, the speed will increase accordingly, so the collected values are positive. Under normal operating conditions, in an ideal state, the acceleration approaches 0 and the speed is at a constant value. However, in the test scenario, the surface of the rail head is in a non - ideal state, so the speed will fluctuate, but the fluctuation is within the normal range and is positive. There are corresponding explanations in the article that the test robot runs the test at a certain constant speed.]
Comments 9:[Additionally, the authors mentioned that the sensor used is a three-axis piezoelectric acceleration sensor. However, the figure does not indicate the direction of measurement. Can the authors specify the orientation of the sensor as shown in the figure? Furthermore, is it possible to extract vibration features from all three directions?]
Response 9:[Thank you very much for the comments from the expert teacher. As shown in Figure 2, the forward direction of the test robot is the X direction, the direction perpendicular to the steel rail is the Y direction, and the vertically upward direction is the Z direction. It has been revised, and the relevant content has been added to the article.Well, it is feasible. ]
Comments 10:[The low operational speed used in this study is literally neither practical nor acceptable. The authors must carefully select experimental parameters that accurately represent real-world conditions when validating their proposed technologies. From this perspective, the Design of Experiments (DoE) is severely inadequate.Moreover, the authors have not accounted for environmental noise and disturbances that may arise at higher operational speeds. The validation conducted at the current test speed lacks rigor and fails to adequately demonstrate the feasibility of defect detection based on the robot’s motion posture.]
Response 10:[Thank you very much for the comments from the expert teacher. Please forgive us. Mainly due to the limitations of the experimental environment, the rail used for testing is 3 meters long. Therefore, when designing the test, the speed was not set very high. However, the data collected at this speed indicates that the proposed method is feasible for the identification and detection of surface damages on the rail head, and the proposed method can complete the identification and detection of surface damages on the rail head.
At the same time, the design of the rail robot is carried out in accordance with the requirements of on - site line detection. This includes the determination of the drive motor model and the design of the transmission mechanism, all of which are designed to complete the detection task on the on - site line. It is mainly restricted by the experimental environment. Meanwhile, we have reached an agreement with CRRC Jiangmen in Guangdong. We will conduct joint line tests on the section of Guangzhou South Station of Guangzhou Railway Bureau and on - site line tests on the Chengdu Passenger Transport Section of Chengdu Railway Bureau. If possible, we sincerely invite you, expert teacher, to come to the site for guidance. Thank you very much for your valuable suggestions and comments!]
Reviewer 2 Report
Comments and Suggestions for Authors
Your answer is somewhat insufficient, but I recommend you insert additional information as a direction for future research. In other words, please clearly state the assumptions and limitations of your research results in your paper.
Author Response
Comments 1: [Your answer is somewhat insufficient, but I recommend you insert additional information as a direction for future research. In other words, please clearly state the assumptions and limitations of your research results in your paper.]
Response 1: [I am extremely grateful for the suggestions from the expert teacher. The limitations of the research results, as well as the key points and directions of future research, have been added and modified in the conclusion section.I am truly thankful for the guidance and opinions provided by the expert teacher.]